# Graph-Based Attention for Differentiable MaxSAT Solving

**Sota Moriyama**[1,2]    **Katsumi Inoue**[2]
[1]The Graduate University for Advanced Studies, SOKENDAI
[2]National Institute of Informatics
{sotam,inoue}@nii.ac.jp

## Abstract

The use of deep learning to solve fundamental AI problems such as Boolean Satisfiability (SAT) has been explored recently to develop robust and scalable reasoning systems. This work advances such neural-based reasoning approaches by developing a new Graph Neural Network (GNN) to differentiably solve (weighted) Maximum Satisfiability (MaxSAT). To this end, we propose SAT-based Graph Attention Networks (SGATs) as novel GNNs that are built on t-norm based attention and message passing mechanisms, and structurally designed to approximate greedy distributed local search. To demonstrate the effectiveness of our model, we develop a local search solver that uses SGATs to continuously solve any given MaxSAT problem. Experiments on (weighted) MaxSAT benchmark datasets demonstrate that SGATs significantly outperform existing neural-based architectures, and achieve state-of-the-art performance among continuous approaches, highlighting the strength of the proposed model.[1]

## 1 Introduction

Neuro-symbolic AI aims to develop robust and scalable reasoning systems by combining the strengths of both symbolic logic and neural networks [20, 6]. Boolean Satisfiability (SAT), a fundamental reasoning problem in AI and Computer Science, has long been examined as an important topic in neuro-symbolic research. There have been many studies on developing SAT solving methods using deep learning, which can either solve SAT problems in an end-to-end manner [22, 1], or can be combined with existing discrete solvers to enhance their performance [21, 11].

Maximum Satisfiability (MaxSAT), an optimization generalization of SAT, has been viewed as a promising approach towards accomplishing various neuro-symbolic tasks [9, 28]. SATNet is one representative approach, capable of learning to solve structured reasoning tasks such as visual sudoku by employing a differentiable MaxSAT solving layer [27]. This layer is built upon a continuous optimization algorithm, specifically based on the use of Semidefinite Programming (SDP) [26]. While this algorithm has proved effective on certain problem types, other promising approaches have also been proposed, such as the use of Fourier analysis [12] and Graph Neural Networks [17]. The latter, however, remains largely underexplored.

Graph Neural Networks (GNNs) have been widely used in relational and symbolic domains, as well as in multiple neuro-symbolic systems [13]. There has recently been a surge of interest in using GNNs as a key building block for combinatorial optimization problems [3]. In particular, several works have applied GNNs to MaxSAT solving, either in an end-to-end manner [17], or by using their predictions as heuristics for existing solvers [16]. In contrast, we propose GNN architectures capable

---

[1]All used code is contained in the repository: https://github.com/sotam2369/SGAT-MS

39th Conference on Neural Information Processing Systems (NeurIPS 2025).

of differentiably solving weighted MaxSAT problems, and are effective against practical problem instances.

In this paper, we present SAT-based Graph Attention Networks (SGATs) as novel GNNs crafted for solving MaxSAT, which are the first GNNs to be able to handle weighted MaxSAT problems. SGATs are composed of GNN layers with novel t-norm based attention, where attention mechanisms operate on values computed using t-norms. When a clause is unsatisfied by a candidate assignment, the clause node sends messages to all its connected variable nodes, requesting them to update their values in parallel towards satisfying the clause. The attention mechanism then computes priorities to decide which variable nodes should change their values to maximally satisfy those requests from clauses, allowing the model to learn which clauses to focus on. Intuitively, this can be regarded as learning a general heuristic that can be applied to a variety of MaxSAT problems. To show the effectiveness of SGATs, we further build a local search algorithm that finds solutions with the continuous optimization of GNNs through iterative training, which can be used with any GNN architecture as the backbone.

To evaluate the effectiveness of our model, we first compare the performance against existing neural-based architectures using benchmark instances from the MaxSAT Evaluations in 2018. We then conduct ablation studies to analyze the effectiveness of each component of SGATs, including the t-norm based attention and message passing mechanisms. Subsequently, we use benchmark instances used in MaxSAT evaluations from 2020 to 2024, and compare the qualities of the predictions with solvers based on state-of-the-art (SOTA) continuous approaches. The experimental results show that SGATs demonstrate excellent stability and performance compared to existing neural architectures and mechanisms during training, and is able to achieve the SOTA performance for continuous solvers on all benchmark sets. We discuss the limitations and broader impacts of our work in Appendix F and Appendix G, respectively.

**Our key contributions are as follows:**

- We present SAT-based Graph Attention Networks (SGATs) as the first GNN architecture specifically designed for MaxSAT solving, and is able to handle weighted MaxSAT problems.
- We introduce novel t-norm based attention and message passing mechanisms that are specifically designed to approximate greedy distributed local search.
- We demonstrate that SGATs outperform existing neural architectures, and achieve SOTA performance for continuous solvers on all benchmark sets.

## 2 Related Work

### 2.1 Differentiable Solvers

In SAT, there have been multiple works attempting to construct differentiable solvers, with deep learning being the main method to accomplish this. Particularly, most works have focused on using GNNs [1, 11] as well as Recurrent Neural Networks (RNNs) [22]. Others have attempted to use these models as heuristics for state-of-the-art (SOTA) solvers, in an effort to further bridge the gap between the two fields [21, 31], with reinforcement learning being one prominent approach [8, 11]. To support further research in this domain, [15] has built a codebase that deploys a wide range of neural architectures, as well as benchmarks known to date. Constraint Satisfaction Problems (CSPs) are another related field, with works focusing on learning search heuristics or end-to-end solvers via neural architectures, with the use of GNNs and transformer variants [23, 29].

In the context of MaxSAT solving, not many differentiable solvers have been proposed to this day, and works that do mainly focus on solving synthesized problems with the use of GNNs [17]. However, there are no works that utilize Graph Attention Networks (GATs), which had been shown to be effective for related problems such as SAT, CSP, and Minimal Unsatisfiable Subset extraction [4, 29, 18]. In contrast, our work focuses on developing a much more robust GNN architecture that employs attention and message passing mechanisms that utilize t-norm computation, a direction not explored in prior works to the best of our knowledge.

## 2.2 Continuous Optimization Based MaxSAT Solvers

Several approaches have been proposed for MaxSAT solving with the use of continuous optimization. The Mixing method [26] is one such approach, specifically using low-rank coordinate descent for Semidefinite Programming (SDP). Together with multiple techniques such as branch-and-bound, it has achieved SOTA performance in solving Max2SAT problems, a special case of MaxSAT problems where each clause has strictly 2 literals [25]. FourierSAT is another continuous optimization based approach, utilizing the Fourier analysis of Boolean functions to handle various types of Boolean constraints [12]. Although this is the only line of work to address weighted MaxSAT, their current implementation does not fully support this, indicating the practical difficulties of handling them.

# 3 Background

## 3.1 SAT, MaxSAT and Weighted MaxSAT

In Boolean Satisfiability (SAT), a propositional logic formula consisting of variables, negations ($\neg$), conjunctions ($\wedge$), and disjunctions ($\vee$) is encoded into Conjunctive Normal Form (CNF). A CNF formula is composed by conjunctions of multiple sub-formulas called *clauses*, with each clause being composed by disjunctions of variables or their negations, called *literals*. Each variable can be assigned a logical value of false (0) or true (1), and the formula is satisfied if and only if there exists an assignment where at least one literal in each clause is mapped to true (each clause is satisfied).

In Maximum Satisfiability (MaxSAT), the objective is to find the assignment of variables that maximizes the number of satisfied clauses in the formula. In weighted MaxSAT, the objective shifts to finding the assignment that maximizes the total weight of satisfied clauses. In the following sections, we will use $n$ and $m$ to denote the number of variables and clauses in a problem, and $w_i$ to denote the weight of clause $C_i$. Furthermore, we refer to cost as the total sum of weights of unsatisfied clauses.

In MaxSAT, there are two major categories of algorithms: complete and incomplete. Complete algorithms guarantee that given solutions are optimal, while incomplete algorithms aim to find good-quality assignments within a reasonable time frame. In our work, we focus on the latter, mainly as guaranteeing optimalities of solutions are difficult with continuous approaches. The current SOTA incomplete solvers are based on Stochastic Local Search, with multiple of them being proposed in recent years [5, 33].

## 3.2 Graph Attention Network

Recent research has shown the usage of GNNs with combinatorial optimization problems (including MaxSAT) to be promising [3, 17]. As such, we take inspiration from Graph Attention Networks (GATs), one of the SOTA architectures for graph based learning [24]. GATs are equipped with attention mechanisms that compute specific weights (attention coefficients) for each neighboring node in order to prioritize important nodes, enabling a leap in model capacity.

The attention mechanism of GATs proceeds as follows. First, the attention coefficients $\epsilon_{ij}$ and normalized attention coefficients $\alpha_{ij}$ for neighbor node $j$ of node $i$ are calculated:

$$\begin{aligned} \epsilon_{ij} &= \text{LeakyReLU}\left(\mathbf{a}^T\left[\mathbf{W}h_i\|\mathbf{W}h_j\|\mathbf{W}_e e_{ij}\right]\right) \\ \alpha_{ij} &= \frac{\exp\left(\epsilon_{ij}\right)}{\sum_{k\in\mathcal{N}_i}\exp\left(\epsilon_{ik}\right)} \end{aligned} \tag{1}$$

Here, $\mathbf{a}\in\mathbb{R}^{2F'}$ is a weight vector, $\|$ is a concatenation operation, and $\mathbf{W}\in\mathbb{R}^{F'\times F}, \mathbf{W}_e\in\mathbb{R}^{F'\times F_e}$ are weight matrices ($F$ and $F'$ represent the number of features in a node at the input and output, and nd $F_e$ the number of features in an edge), with the LeakyReLU nonlinearity being applied (ReLU with a slight negative slope). Furthermore, $h_i\in\mathbb{R}^F$ is the feature of node $i$, $e_{ij}\in\mathbb{R}^{F_e}$ is the edge feature between node $i$ and $j$, and $\mathcal{N}_i$ is the set of neighbors of node $i$. The above notation is slightly modified to allow the use of multi-dimensional edge features [7]. Subsequently, these normalized attention coefficients are used to compute the final output feature $h_i'\in\mathbb{R}^{F'}$ for node $i$:

$$h_i' = \sum_{j\in\mathcal{N}_i}\alpha_{ij}\mathbf{W}h_j \tag{2}$$

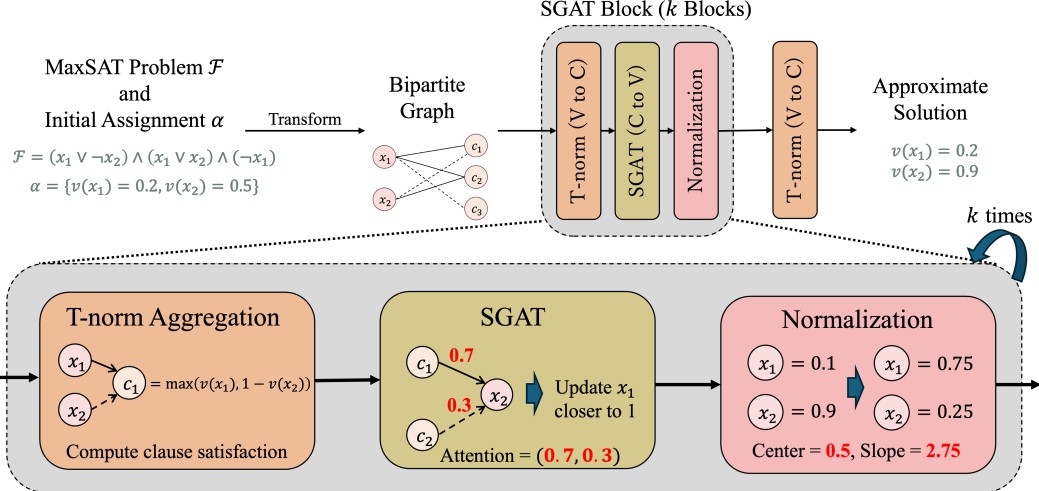

Figure 1: Architectural Diagram of SGATs. Each SGAT block is composed of a t-norm layer, SGAT layer, and a normalization layer. Red represents the values that are learned.

# 4 SAT-based Graph Attention Network

In this section, we present key building blocks for constructing SGATs as shown in Figure 1. SGATs are composed of three main components: (i) T-norm layers that compute clause valuations $v(C)$ from the valuations of connected variables $v(x)$, (ii) SGAT layers that update the valuations of variables based on the valuations of connected clauses, and (iii) a normalization layer that ensures the valuations of all variables are sufficiently spread out. A single block of SGATs approximates a distributed local search step, with the attention mechanism learning which clauses to focus on, essentially serving as a heuristic. We denote variables by $x_i$ and clauses by $C_j$, with valuations $v(x_i), v(C_j) \in [0, 1]$. For clarity, $v(x_i)$ and $v(C_j)$ correspond to features at nodes $h_i$ and $h_{j+n}$, respectively.

## 4.1 Graph Representation of MaxSAT Problems

Before applying GNNs to MaxSAT problems, we have to transform them into graphs. To accomplish this, we employ a factor graph representation of given formulas, as done in most prior works involving SAT and MaxSAT Solving [22, 30, 17]. Specifically, we opt for a representation similar to [30], and obtain a bipartite graph with two types of nodes for both the clauses and variables as shown in Figure 2. The positive and negative polarities of the variables are then embedded into the edge features.

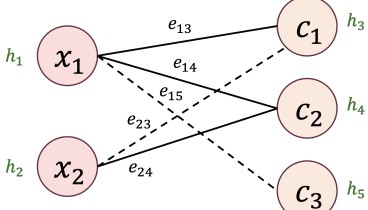

Figure 2: Bipartite graph for $\mathcal{F} = (x_1 \vee \neg x_2) \wedge (x_1 \vee x_2) \wedge (\neg x_1)$. Solid and dashed lines each represent the positive and negative polarities respectively.

While existing graph representations used edge features to mainly differentiate between the polarities of variables, we propose to embed crucial information as edge features, to allow the model to further differentiate between similar connections. We define the edge feature $e_{i,j+n}$ between variable $x_i$ and clause $C_j$ as follows:

$$e_{i,j+n} = \begin{cases} \left( \frac{1}{|\mathcal{N}_j^+|}, \frac{1}{|\mathcal{N}_j^-|}, 0, 0 \right) \cdot w_j^{\mathrm{norm}} & \text{if } i \in \mathcal{N}_j^+ \\ \left( 0, 0, \frac{1}{|\mathcal{N}_j^+|}, \frac{1}{|\mathcal{N}_j^-|} \right) \cdot w_j^{\mathrm{norm}} & \text{if } i \in \mathcal{N}_j^- \end{cases}, \quad \text{where } w_j^{\mathrm{norm}} = \frac{w_j}{\max_k w_k}$$

Here, $\mathcal{N}_j^+$ and $\mathcal{N}_j^-$ correspond to the set of variables with positive and negative polarities included in clause $C_j$, and $w_j^{\mathrm{norm}} \in (0, 1]$ represents the normalized weight of clause $C_j$. Note that when the denominator is 0, we set the corresponding value to 0. These features align with common strategies

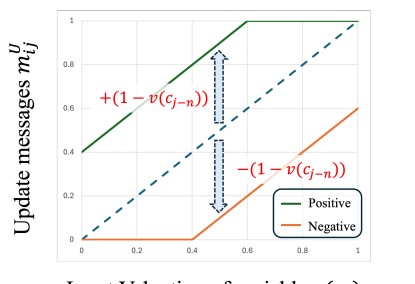
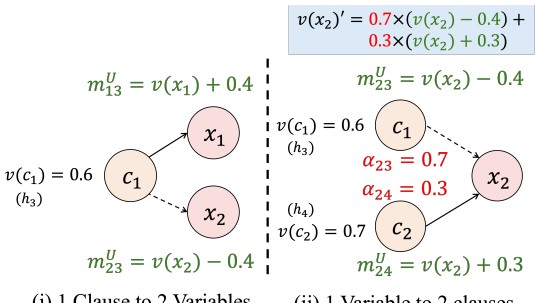

Input Valuation of variable $v(x_i)$     (i) 1 Clause to 2 Variables     (ii) 1 Variable to 2 clauses

(a) Update message $m_{ij}^U$ for $x_i$ and $C_{j-n}$. The above is a visualization for $v(C_{j-n}) = 0.6$.

(b) Visualization of the update procedure with SGATs.

Figure 3: Illustration of the SGAT layer.

prioritizing shorter clauses over long ones [10], while also giving higher scores to clauses with larger weights; this is especially relevant as edge features are directly used for attention computation.

## 4.2 T-norm Layer: Variable to Clause

Here, we propose the use of t-norm based aggregations when computing clause valuations from connected variable valuations, as shown in the bottom left of Figure 1. By using t-norms, we compute clause valuations continuously while keeping all valuations in $[0, 1]$. Given fuzzy truth degrees $a_1, \ldots, a_k \in [0, 1]$ the fuzzy conjunctions of these values can be computed using t-norms $T : [0, 1]^k \to [0, 1]$. We specifically use the following three well-known t-norms:

1. Gödel t-norm: $T_G(a_1, \ldots, a_k) = \min(a_1, \ldots, a_k)$
2. Product t-norm: $T_P(a_1, \ldots, a_k) = \prod_{i=1}^{k} a_i$
3. Łukasiewicz t-norm: $T_L(a_1, \ldots, a_k) = \max\left(0, \sum_{i=1}^{k} a_i - (k-1)\right)$

As each clause is a disjunction of literals, we compute valuations by negating the conjunction of each negated literal. Using strong negation defined as $1 - a$ for $a \in [0, 1]$, we set $\tilde{v}_{ir} = 1 - v(x_{ir})$ if $l_{ir} = x_{ir}$ and $\tilde{v}_{ir} = v(x_{ir})$ if $l_{ir} = \neg x_{ir}$ for a clause $C_i = l_{i1} \vee \cdots \vee l_{i\ell_i}$, and define

$$v_\star(C_i) = 1 - T_\star(\tilde{v}_{i1}, \ldots, \tilde{v}_{i\ell_i}), \qquad \text{where } \star \in \{G, P, L\}.$$

For simplicity, we refer to the clause valuation computed by the chosen t-norm as $v(C_i)$.

**Unsupervised Loss Function.** By applying a t-norm aggregation, we obtain clause valuations $v(C_k)$. We then define a loss function that computes the total cost of the current assignment as:

$$\mathcal{L} = \frac{\sum_{k=1}^{m} w_k \left(1 - v(C_k)\right)^2}{\sum_{k=1}^{m} w_k} \tag{3}$$

where $m$ is the total number of clauses. The equation is a weighted mean squared error of how unsatisfied each clause is, with the weights corresponding to the given weights for each clause. This loss function forces the model to train to minimize the total weight of unsatisfied clauses, without requiring any sort of ground truth labels, with the assumption that a cost of 0 (all clauses satisfied) is the optimal solution.

## 4.3 SGAT Layer: Clause to Variable

To efficiently learn which variables to update, we employ an attention mechanism and a message-passing function that update the valuations of variables based on the valuations of the clauses to which they are connected (Figure 3b). Specifically, we define the update message $m_{ij}^U$ and attention message $m_{ij}^A$ between variable $x_i$ and clause $C_{j-n}$ as follows:

$$m_{ij}^U = \begin{cases} v(x_i) + \left(1 - v(C_{j-n})\right) & \text{if } i \in \mathcal{N}_{j-n}^+ \\ v(x_i) - \left(1 - v(C_{j-n})\right) & \text{otherwise} \end{cases}, \qquad m_{ij}^A = \begin{cases} m_{ij}^U & \text{if } i \in \mathcal{N}_{j-n}^+ \\ 1 - m_{ij}^U & \text{otherwise} \end{cases}$$

The update messages $m_{ij}^U$ represent the valuation the clause requests the variable to move toward, as shown in Figure 3a. The attention messages $m_{ij}^A$ represent the strength of this request by flipping the valuation for negatively appearing literals, aligning all messages in a common direction regardless of polarity. Using these messages, Equations (1) and (2) are redefined as:

$$\epsilon_{ij} = \text{LeakyReLU}\left(\mathbf{a}^T \left[\mathbf{W}m_{ij}^A \| \mathbf{W}_e e_{ij}\right]\right)$$

$$v'(x_i) = \sum_{j \in \mathcal{N}_i} \alpha_{ij} m_{ij}^U$$

From the equation, as long as the input valuations $v(x_i) \in [0, 1]$, $\alpha_{ij} \in [0, 1]$, and $\sum_{j \in \mathcal{N}_i} \alpha_{ij} = 1$, the outputs $v'(x_i) \in [0, 1]$ are guaranteed to hold. This eliminates the need for additional activation functions after each layer, simplifying the computation of both variable assignments and the loss function. Additionally, we clamp the values of update messages to range $[0, 1]$ whenever they leave the range (possible with certain t-norms). Extensive information on how SGATs handle input-output dimensions is provided in Appendix A.

SGATs are approximations of greedy distributed local search; they perform local search in a fully parallel and greedy manner. After computing clause valuations $v(C)$, they send messages $(m_{ij}^U)$ to all connected variable nodes (greedy) in parallel, with the strength proportional to how unsatisfied the clause valuation remains $(m_{ij}^A)$. The variables then use the attention mechanism explained above to decide which clause messages to prioritize, to maximally satisfy those requests from clauses. SGATs learn these distributed local search heuristics through attention, which lead to the best approximations.

**Multi-head attention.** Velickovic et al. [24] further employed multi-head attention to stabilize the learning process of self-attention for GATs. We also employ the same strategy by computing $K$ independent attention coefficients and averaging the outputs of all heads. As the outputs for each head are guaranteed to be in range [0,1], the final outputs here are also guaranteed to be in range [0,1].

## 4.4 Normalization Layer

To prevent feature values from being overly concentrated or dispersed, we apply a normalization layer based on the sigmoid function, as shown in the bottom right of Figure 1. The normalized feature for node $h_i' \in [0, 1]$ is computed as:

$$h_i' = \sigma\big(\gamma\left(2h_i - 1 - k\right)\big)$$

where $\sigma(\cdot)$ denotes the sigmoid function, $\gamma$ a parameter for controlling the spread of the values, and $k \in [-1, 1]$ a parameter for controlling the center of the values. The input feature $h_i \in [0, 1]$ is first scaled to the $[-1, 1]$ range, after which the sigmoid modulates how close the output is to 0 or 1.

## 4.5 Approximation Ratio of SGATs

In previous works [17, 25], the capabilities of differentiable MaxSAT solvers were evaluated theoretically using the approximation ratio, which is the lower bound of the ratio of the value computed with the algorithm to the optimal value. In [17], a $1/2$-approximation ratio for unweighted MaxSAT is achieved by GNN architectures with hidden dimension sizes dependent on the number of clauses. We can prove that SGATs achieve a $1/2$-approximation for any unweighted Max-E$k$SAT problem (problems with exactly $k$ literals per clause), and that this guarantee holds with fixed architectural settings independent of the number of clauses. This result establishes a deterministic baseline that helps us understand the theoretical capabilities of SGATs. The details are provided in Appendix E.

## 4.6 Local Search with GNNs

To solve MaxSAT problems with SGATs, we introduce LS-GNN (Local Search with GNN, Algorithm 1), a novel local search solver based on the continuous optimization of SGATs. LS-GNN takes as input a weighted MaxSAT instance $\mathcal{F}$ and returns the best valuation $v_{\text{best}}$ and cost $\mathcal{C}_{\text{best}}$ found for a given problem within a time limit. Starting from a random valuation $v$, SGAT maps the current valuations of variables and clauses to valuations that aim to maximize clause satisfaction. We iteratively optimize SGATs to minimize the loss in Equation (3), moving the valuations toward better solutions.

**Algorithm 1:** LS-GNN
***
**In** : Weighted MaxSAT instance $\mathcal{F}$, timeout $T$.
**Out** : Best valuation $v_{\text{best}}$ and its cost $\mathcal{C}_{\text{best}}$.

1   $v_{\text{best}} = \emptyset$, $\mathcal{C}_{\text{best}} = +\infty$, $v = $ random assignment
2   **while** *elapsed time* $< T$ **do**
3     $k = 0$, $k_{\text{local}} = 0$, $v_{\text{local}} = \emptyset$, $\mathcal{C}_{\text{local}} = +\infty$
4     **while** *early stopping* $\geq k - k_{local}$ **do**
5       optimize SGAT with respect to $v$
6       $\mathcal{C} = \text{cost}\big(\mathcal{F}, \text{SGAT}(\mathcal{F}, v)\big)$
7       **if** $\mathcal{C} < \mathcal{C}_{local}$ **then**
8         $v_{\text{local}} = \text{SGAT}(\mathcal{F}, v)$, $k_{\text{local}} = k$, $\mathcal{C}_{\text{local}} = \mathcal{C}$
9       **end**
10       $k = k + 1$
11     **end**
12     **if** $\mathcal{C}_{local} < \mathcal{C}_{best}$ **then**
13       $v_{\text{best}} = v_{\text{local}}$, $\mathcal{C}_{\text{best}} = \mathcal{C}_{\text{local}}$
14     **end**
15     $v = (1 - \boldsymbol{\beta}) \cdot v + \boldsymbol{\beta} \cdot \Delta$
16   **end**
***

To reduce the risk of the algorithm being stuck in a local optimum, we periodically randomize a subset of valuations (L15). This is done by sampling a binary mask $\beta$ via thresholding a uniform random distribution, and injecting new random valuations $\Delta$ for the masked variables. The randomization procedure is triggered whenever the best cost stops improving for *early stopping* epochs (L4). The parameters *early stopping*, *elapsed time*, $\beta$, and $\Delta$ control the termination and randomization behavior of LS-GNN; their specific values are provided in Appendix A.

While Algorithm 1 used SGATs as the GNN model, it can be used with any arbitrary model that is able to output assignment predictions. For simplicity, we refer to the LS-GNN with SGATs and GATs as LS-SGAT and LS-GAT, respectively.

## 5 Experiments

In this section, we conduct multiple experiments to answer the following questions regarding SGATs: **Q1)** Are SGATs better than existing GNN architectures? **Q2)** What component makes our model efficient? and **Q3)** How can LS-GNN be compared with other continuous solvers? For evaluation, we use instances used in non-partial unweighted and weighted benchmark instances provided in MaxSAT evaluations[2], which we denote as **MS** and **WMS** (and collectively referred to as **WMS+**). We also prepare a subset of these datasets with instances that are below specific file sizes such as 2MB for purposes such as training, and denote as **WMS+**(2MB). The training and testing splits are shown in Appendix D.

### 5.1 Model Architecture

To evaluate the effectiveness of our model in learning to solve practical instances (MaxSAT Evaluation benchmark instances), we first conduct experiments using different model architectures. Provided that there is only one work regarding end-to-end MaxSAT solving with GNNs [17], we also compare with models that were built for SAT. Specifically, we use NeuroSAT [22] and GGNN [14] with unsupervised loss functions presented in [19], as they were shown to empirically work well [15]. Specific details regarding models are given in Appendix B.

We compare the model performance on the **MS2018**(2MB) dataset, with training done on two different datasets: (i) **MS2018**(2MB), and (ii) **SR**(**U**(40,200)), a randomly generated dataset with 40

***
[2]https://maxsat-evaluations.github.io

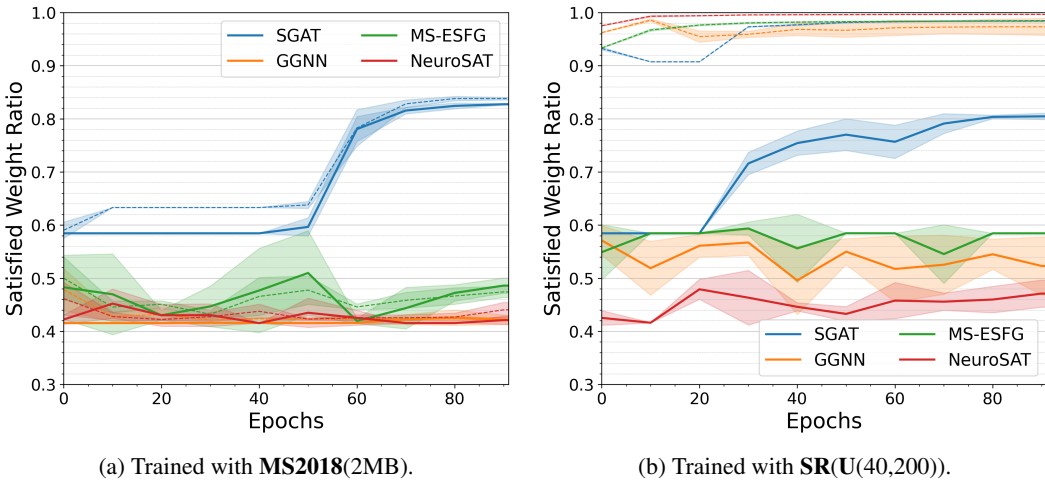

(a) Trained with **MS2018**(2MB).  (b) Trained with **SR**(**U**(40,200)).

Figure 4: Comparative analysis of model architectures, with different training datasets. Test sets for both were **MS2018**(2MB). Solid: Test, Dashed: Train.

to 200 variables [22]. We use the satisfied weight ratio as a performance metric, which is defined as:

$$\text{Satisfied Weight Ratio} = \frac{\sum_{k=1}^{m} w_k \cdot \text{round}\big(v_G(C_k)\big)}{\sum_{k=1}^{m} w_k},$$

where $v_G(C_k) \in [0, 1]$ is the valuation of clause $C_k$ computed using the Gödel t-norm, $w_k$ is the weight of clause $C_k$, and $m$ is the total number of clauses. This score represents the ratio of the total weight of satisfied clauses to the total weight of all clauses, with a higher score indicating a strictly better solution.

We used SGATs with 6 SGAT blocks composed with Gödel T-norm layers and SGAT layers with 2 attention heads and 4 channels. For training, we used the Adam optimizer with a learning rate of $2 \times 10^{-3}$, and a batch size of 4. For existing models, the default provided settings were used. The number of blocks are generally chosen to strike the best balance between performance and computational efficiency, with the specific empirical results shown in Appendix D.

### 5.1.1 Results

As shown in Figure 4, our model is able to learn to solve from practical instances, while others completely fail to do so. Given that the training dataset consists of problems from various domains, we can infer that SGATs are capable of learning to solve a wide range of problems. Figure 4b further supports this conclusion; even with synthesized datasets, our model is able to learn a distributed local search heuristic that has high performance on practical (non-synthesized) datasets. Overall, this shows our models' dominant strength in solving MaxSAT problems. Additionally, the scalability difference between existing models and SGATs is shown to be huge, due to the large difference in parameter numbers. For the training with **MS2018**(2MB), existing architectures could only handle a batch size of 1. In contrast, SGATs could support batch sizes of up to 4, contributing to the high stability of our model. Extensive results are shown and discussed in Appendix D.

### 5.2 Ablation Study

To highlight which components of our model yields the highest performance increase, we performed ablation studies focusing on two points: (a) SGAT and T-norm layer, and (b) types of t-norm. For (a), we consider two variants of our model, one with SGATs swapped to GAT with Sigmoid, and another swapping out the T-norm layers with GATs. As SGATs are dependent on T-norm layers being used, we do not experiment with T-norm layers swapped with SGATs. For (b), we compare using three different fundamental t-norms that have been frequently used for machine learning: Gödel, Product, and Łukasiewicz. The experiments were all performed with **WMS+2018**(2MB) as the training and testing set, with the same model parameters as the previous experiment.

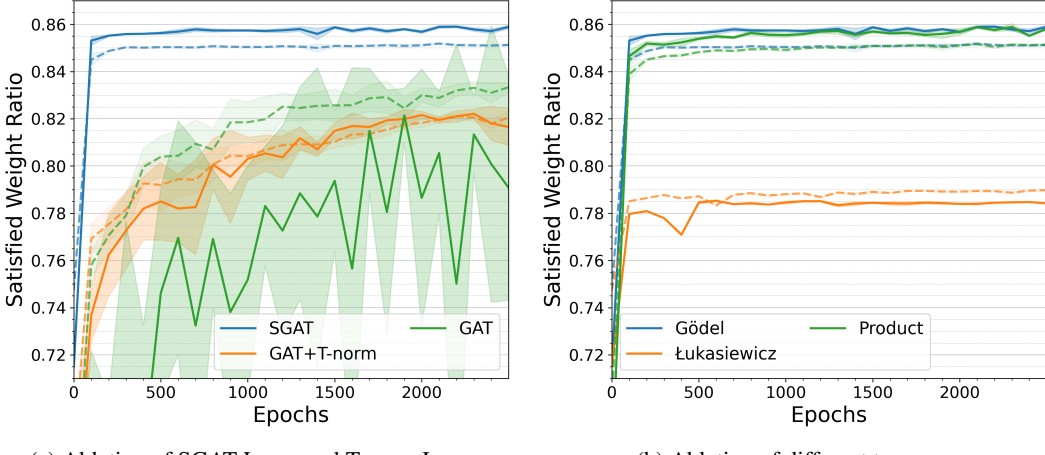

| (a) Ablation of SGAT Layer and T-norm Layer. | (b) Ablation of different t-norms. |

Figure 5: Results of ablation studies. Training and testing were done on **WMS+2018**(2MB). Solid: Test, Dashed: Train.

### 5.2.1 Results

Figure 5 shows the results for each ablation study. From the first ablation study, we can see that our current model architecture achieves the best performance, with significantly high stability. Furthermore, we can see that having the T-norm layer for clause value updates in place of a GAT layer highly increases the stability, even though the total number of trainable layers essentially halve.

From the second ablation study, we can observe that Gödel and Product t-norms work substantially better than Łukasiewicz. This is thought to be because Łukasiewicz outputs 0 when the input values are too low, making the gradients zero in certain regions, hindering the training process. On the other hand, the other two t-norms do not have this issue, resulting in more stable and effective optimization. Additionally, we can observe that Gödel converges slightly faster than Product. This is thought to be due to the compatibility between SGAT layers and Gödel t-norm; as it is guaranteed that no variable will have higher values than the clause it is connected to, the update message never goes out of the range [0,1], removing the need of any clamping procedures.

### 5.3 Local Search with SGATs

In our second experiment, we tested the performance of our solvers based on continuous optimization at approximating solutions on both unweighted and weighted benchmark instances. We used LS-GNN with GATs and SGATs pretrained in the previous experiment (as LS-GAT and LS-SGAT) as backbone models. We compared our solvers to three existing solvers based on continuous optimization: the Mixing method [25], the MIXSAT algorithm [26], and FourierSAT [12]. The default settings were used for all solvers. As the only continuous solver that addresses weighted MaxSAT is FourierSAT, we conduct experiments comparing our model only with FourierSAT on **WMS** instances. We specifically make implementation changes to support weights, as the original does not do so at default.

To evaluate the approximations computed by each solver, we calculated the incomplete scores, a metric that has been used to evaluate incomplete MaxSAT solvers in recent years of MaxSAT evaluations. The incomplete scores are defined as

$$\text{score}(s, i) = \frac{(1 + \text{cost of best known cost for } i)}{(1 + \text{cost of solution for } i \text{ found by } s)},$$

where $i$ is the instance and $s$ is the solver used. For reference, we retrieve the best known cost for each instance from the official MaxSAT Evaluation results. We combine this with the best found cost for each dataset and timeout setting, keeping it comparable with existing methods. However, we note that due to our solvers finding better solutions than the competition results, the best known cost is not completely identical to the official results.

Table 1: Average Incomplete Score for solvers based on continuous optimization, evaluated on unweighted (**MS**) and weighted (**WMS**) benchmark instances. Bold shows the best score, and underlined show the second best score.

| Type | Solvers | 60s timeout | | | | | 300s timeout | | | | |
|---|---|---|---|---|---|---|---|---|---|---|---|
| | | 2020 | 2021 | 2022 | 2023 | 2024 | 2020 | 2021 | 2022 | 2023 | 2024 |
| **MS** | Mixing | 0.3729 | 0.3442 | 0.3864 | 0.0703 | 0.1564 | 0.3603 | 0.3443 | 0.3864 | 0.0703 | 0.1564 |
| | MIXSAT | 0.4887 | 0.5017 | 0.4980 | 0.1367 | 0.3077 | 0.4796 | 0.5053 | 0.5269 | 0.1429 | 0.3100 |
| | FourierSAT | 0.4304 | 0.4023 | 0.3560 | 0.1429 | 0.1189 | 0.4498 | 0.4672 | 0.4205 | 0.1692 | 0.1595 |
| | LS-GAT | 0.3948 | 0.4268 | 0.4839 | 0.0331 | 0.1284 | 0.4281 | 0.4359 | 0.5263 | 0.0919 | 0.1413 |
| | LS-SGAT | **0.5237** | **0.5420** | **0.6202** | **0.1653** | **0.3516** | **0.5289** | **0.5541** | **0.6403** | **0.1828** | **0.3684** |
| **WMS** | FourierSAT | 0.2304 | 0.3340 | 0.4452 | 0.2974 | 0.0144 | 0.2369 | 0.3523 | 0.4490 | 0.2967 | 0.0163 |
| | LS-GAT | 0.6936 | 0.7197 | 0.6925 | 0.7636 | 0.6650 | 0.7384 | 0.8005 | 0.7283 | 0.8216 | 0.7156 |
| | LS-SGAT | **0.8499** | **0.8973** | **0.7587** | **0.8432** | **0.7816** | **0.8342** | **0.8835** | **0.7594** | **0.8471** | **0.7840** |

### 5.3.1 Results

Table 1 shows the average incomplete score for each solver depending on the year of evaluation. From the results, we can see that LS-SGAT clearly outperforms every other existing solver based on continuous optimization, with an average improvement of 0.055 for **MS**, and 0.09 for **WMS** versus the second best solver. However, we can also observe that the differences are not constant for every year. This is analyzed to be due to the types of problem each year contains; some solvers perform better on specific instances than others. Nevertheless, we can confirm that the overall, SGATs perform significantly better on a wide range of problems compared to existing continuous methods.

Another important factor that we observed was that the size of problems the algorithm was able to handle was much different. While SGATs were able to handle all but one instance, others went over the timelimit on tens to up to hundreds of instances, showing that our model and algorithm scales much better than existing approaches. This is mainly due to the efficiency of GNN architectures; GNNs tend to have much fewer parameters, which are independent of the problem size. We further analyze this with more experimental results in Appendix D.

**SGATs as Initialization Heuristics.** While our primary concern is to develop a differentiable method for MaxSAT solving, it is worth investigating how SGATs can predict good assignments for a given Weighted MaxSAT instance and whether such a predicted assignment can be used as an initial assignment for a SOTA solver. In this context, we have conducted an additional experiment where the predictions of SGATs are used as initialization heuristics for state-of-the-art incomplete solvers. The results were positive, with incomplete solvers being able to achieve substantially higher incomplete scores when combined with SGATs, supporting our claim that SGATs can work equally well in practical settings. The extensive results are shown in Appendix C.

## 6 Conclusion and Future Work

We presented SGATs as novel GNNs that utilize attention and message passing mechanisms that operate on t-norms. SGAT layers approximate greedy distributed local search, with their heuristics being the main learnable component. To demonstrate the effectiveness of our model, we further developed a continuous local search algorithm that is built on top of SGATs to solve given MaxSAT instances in a continuous manner. Experimental results showed that SGATs train in a highly stable manner, with approximations clearly outperforming those produced by existing neural-based architectures. Our model also outperforms state-of-the-art continuous solving approaches, demonstrating the strength of our model to output good approximations, even against theoretically sound approaches.

Future works will be focused on expanding the current framework to support partial MaxSAT problems, which require the satisfaction of hard clauses, and is known to be difficult for Neural Networks. Another interesting direction would be to incorporate SGATs into machine learning systems for tasks such as recognition and prediction. Especially in neuro-symbolic systems where MaxSAT solvers are used [9], SGATs can easily replace the solvers to allow for end-to-end learning.

## Acknowledgments

This work has been supported by JSPS KAKENHI Grant Number JP25K03190, JST CREST Grant Number JPMJCR22D3 and JST SPRING Grant Number JPMJSP2104, Japan.

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

# A  Additional Details on SGATs

## A.1  SGAT Architecture

The SGAT model begins with a lightweight initializer module that transforms the input variable assignment into a hidden embedding. This module is implemented as a single-layer MLP:

$$\texttt{init}(x) = \sigma(\mathbf{W} \cdot x + b)$$

where $x \in \mathbb{R}^{n \times 1}$ is the initial assignment (usually $x_i \sim \text{Uniform}(0, 1)$), $\sigma$ is a sigmoid activation, and $\mathbf{W}$ is a learned weight matrix mapping from dimension 1 to the hidden dimension $d$.

The purpose of the initializer is to project scalar input values into a richer embedding space before message passing begins. This allows the attention mechanism in subsequent SGAT blocks to operate over informative feature vectors rather than raw values.

Empirically, this step contributes to faster convergence and higher clause satisfaction, especially during early training. The initializer is shared across all variables and does not depend on graph structure, ensuring a consistent embedding style for all inputs.

On the other hand, outputs are simply averaged, essentially making the final assignment of variables a vote over all hidden dimensions.

## A.2  Early Stopping and Random Restarts for LS-GNN

During the search we combine per-instance early stopping with controlled variable perturbation and occasional random restarts to avoid local optima. Each instance maintains its own early-stopping counter and patience, which is adaptively increased when significant improvements are observed; when the counter triggers, we do not fully reinitialize the assignment. Instead a binary mask $\beta$ is sampled by thresholding a uniform distribution at per-instance probability $p$ (the probabilities cycle through a schedule such as $\{0.1, 0.2, 0.3, 0.5, 1.0\}$ on successive triggers). New random values $\Delta$ are sampled and used to replace only the subset of variables selected by $\beta$, preserving useful partial assignments while injecting diversity. Occasionally, when prolonged stagnation occurs, we perform a full random restart. This adaptive combination of targeted perturbation and occasional restarts balances local exploitation and global exploration, improving the solver's ability to escape local minima while retaining promising partial solutions.

# B  GNN Architectures

Table 2: Training configurations for GNN baselines. All models were trained for 100 epochs with early stopping and learning rate scheduling. The MS-ESFG configuration was manually added for comparison.

| Model | Graph | Iterations | Hidden Dim | LR | Weight Decay |
|-------|-------|-----------|-----------|-----|--------------|
| GGNN [14] | VCG | 32 | 64 | 0.002 | $1 \times 10^{-8}$ |
| NeuroSAT [22] | LCG | 32 | 128 | 0.002 | $1 \times 10^{-8}$ |
| MS-ESFG [17] | VCG | 20 | 128 | $2 \times 10^{-5}$ | $1 \times 10^{-10}$ |

**GGNN** [14]: A gated graph neural network architecture that uses GRU-style updates to propagate information over graph nodes. It has been commonly used for reasoning tasks due to its recurrent structure.

**NeuroSAT** [22]: A message passing neural network designed for satisfiability problems, using symmetric updates over literals and clauses on the LCG (literal-clause graph).

**MS-ESFG** [17]: A GNN architecture proposed by Liu et al. [17] for learning to solve MaxSAT. It uses edge-splitting factor graphs (ESFG) with a transformer-style encoder and was evaluated on synthetic MaxSAT benchmarks.

**VCG and LCG** VCG (Variable-Clause Graph) is a bipartite graph representation where nodes correspond to variables and clauses, with edges indicating the inclusion of a variable (or its negation)

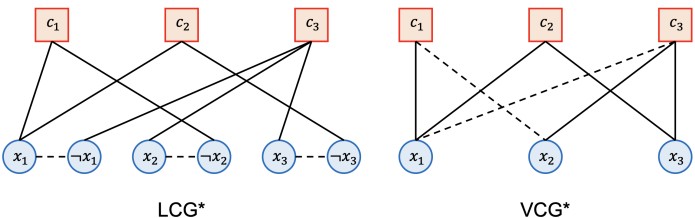

Figure 6: Visual comparison between VCG and LCG representations, as shown in [15].

in a clause. Polarity (positive or negative) is often encoded as edge features. LCG (Literal-Clause Graph), used in NeuroSAT, instead treats each literal (i.e., $x_i$ and $\neg x_i$) as a separate node, resulting in a more fine-grained graph with symmetric updates between literal and clause nodes (see Figure 6).

To implement all GNNs above, including GGNN, NeuroSAT, and MS-ESFG, we utilized the open-source G4SATBench framework [15], which provides standardized model definitions, training procedures, and evaluation protocols for GNN-based SAT solvers. This ensures consistency across implementations and allows for fair comparison under unified training and evaluation settings. Additionally, we modified the evaluation routine to calculate Clause Satisfaction, as defined and used in the main paper shown in Section 5.1.

### B.1 Unsupervised Loss Function

All models were trained using the unsupervised loss function originally proposed by Ozolins et al. [19]. Let $v(x_i) \in [0, 1]$ denote the valuation assigned to variable $x_i$. The loss is defined as

$$\mathcal{L}_\phi(v) = -\sum_{C \in \phi} \log \left( 1 - \prod_{x_i \in C^+} \left(1 - v(x_i)\right) \prod_{x_i \in C^-} v(x_i) \right), \tag{4}$$

where $C^+$ and $C^-$ denote the sets of variables appearing positively and negatively in clause $C$, respectively. This formulation provides a smooth and fully differentiable estimate of clause valuations, allowing effective training without ground-truth labels.

---

**Algorithm 2:** SLS solver with SGAT-Based initialization

**Input** : MaxSAT instance $\mathcal{F}$, timeout $T$.
**Output :** Best solution found $x_{\text{best}}$ and its cost $C_{\text{best}}$.

1   $x_{\text{best}} = \emptyset$, $C_{\text{best}} = +\infty$
2   **while** *elapsed time* $< T$ **do**
3     $x = \texttt{SGAT\_Initialize}(\mathcal{F})$
4     **while** *Restart condition not reached* **do**
5       $C = \texttt{cost}(x)$
6       **if** $C < C_{best}$ **then**
7         $x_{\text{best}} = x$, $C_{\text{best}} = C$
8       **end**
9       $l = \texttt{SelectVariable}()$
10      $\texttt{Flip}(x, l)$
11     **end**
12   **end**
13   **return** $x_{\text{best}}$, $C_{\text{best}}$

---

---

**Algorithm 3:** SGAT-Based Initialization

---

**Input** : MaxSAT instance $\mathcal{F}$.
**Output** : Initial assignment $x$.

1  $v(x)^0 = \text{RandomAssignment}()$
2  $v(x)^{\text{SGAT}} = \text{SGAT}(\mathcal{F}, v(x)^0)$
3  **foreach** *variable $x_i$ in $\mathcal{F}$* **do**
4      **if** $Uniform(0, 1) < v(x)_i^{SGAT}$ **then**
5          $x_i = 1$
6      **else**
7          $x_i = 0$
8      **end**
9  **end**
10 **return** $x$

---

## C    SGATs as Initialization Heuristics

### C.1    Stochastic Local Search with SGAT

We propose SLS-SGAT (Stochastic Local Search with SGAT), a solver that integrates SGAT predictions as initialization heuristics for state-of-the-art SLS solvers, as shown in Algorithms 2 and 3. This builds on prior work that uses GNN predictions for heuristic guidance in SAT solvers [31], but introduces a novel variable assignment scheme based entirely on SGAT predictions. Specifically, each variable is initialized based on thresholding its predicted value, enabling the solver to take full advantage of the inductive biases learned by SGATs.

In principle, this approach can be applied to any SLS solver that relies on random initialization. In this paper, we evaluate this technique by modifying four SOTA solvers to follow the SGAT-based initialization scheme shown in Algorithm 2.

### C.2    Experiments

In this experiment, we evaluate whether SGAT predictions can serve as effective initialization heuristics for four state-of-the-art SLS solvers: SPB [33], NuWLS [5], BandHS [32], and SATLike3.0 [2]. These solvers represent the top-performing incomplete MaxSAT solvers from recent MaxSAT Evaluations, with NuWLS and SPB featured in winning entries from 2022 to 2024.

Each solver was modified to incorporate the SGAT-based initialization strategy described in Algorithms 2 and 3. To ensure a fair comparison, SGAT inference time was included in the overall solver timeout.

Table 3: Average Incomplete Score for each SLS solver and our proposed modification, evaluated on *unweighted and weighted* benchmark instances. Bold shows the best score, and underlined shows the second best score.

| Solvers | 60s timeout | | | | | 300s timeout | | | | |
|---|---|---|---|---|---|---|---|---|---|---|
| | 2020 | 2021 | 2022 | 2023 | 2024 | 2020 | 2021 | 2022 | 2023 | 2024 |
| SPB | 0.8964 | 0.9175 | 0.9159 | 0.8862 | 0.7699 | 0.9050 | 0.9206 | 0.9215 | 0.8907 | 0.7820 |
| + SLS-SGAT | 0.9045 | **0.9220** | 0.9200 | **0.8955** | 0.7845 | **0.9178** | 0.9258 | **0.9335** | **0.8998** | 0.7914 |
| NuWLS | 0.8932 | 0.9143 | 0.9155 | 0.8807 | 0.8885 | 0.9045 | 0.9213 | 0.9177 | 0.8907 | 0.9053 |
| + SLS-SGAT | **0.9049** | 0.9209 | **0.9217** | 0.8944 | **0.9036** | 0.9175 | **0.9280** | 0.9327 | 0.8988 | **0.9134** |
| BandHS | 0.8563 | 0.8754 | 0.8630 | 0.8696 | 0.7412 | 0.8751 | 0.8903 | 0.8841 | 0.8810 | 0.7591 |
| + SLS-SGAT | 0.8616 | 0.8813 | 0.8634 | 0.8783 | 0.7557 | 0.8785 | 0.8924 | 0.8827 | 0.8888 | 0.7747 |
| SATLike3.0 | 0.8327 | 0.8474 | 0.8193 | 0.8582 | 0.7393 | 0.8479 | 0.8628 | 0.8295 | 0.8718 | 0.7543 |
| + SLS-SGAT | 0.8388 | 0.8640 | 0.8093 | 0.8707 | 0.7492 | 0.8521 | 0.8754 | 0.8367 | 0.8781 | 0.7671 |

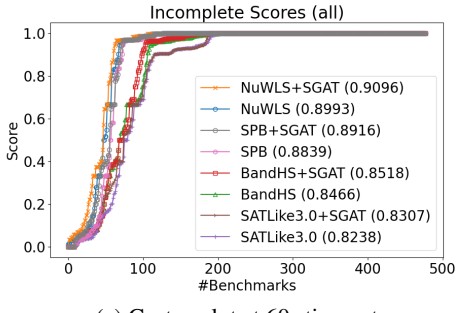
(a) Cactus plot at 60s timeout.

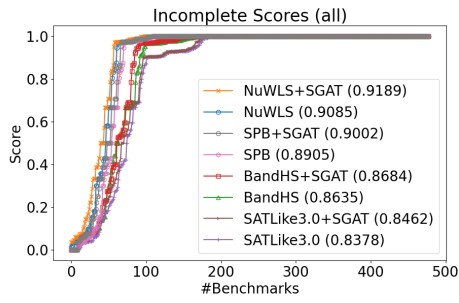
(b) Cactus plot at 300s timeout.

Figure 7: Cactus plots comparing incomplete solvers before and after SGAT-based initialization. Y-axis shows the score achieved, and X-axis shows the number of benchmark instances. Higher is better.

### C.2.1 Results

Table 3 presents the average incomplete scores across benchmark sets from 2020 to 2024. All solvers consistently benefited from SGAT-based initialization, with hybrid versions of NuWLS and SPB achieving state-of-the-art results. Notably, NuWLS showed an average improvement of 0.008—comparable to or greater than the score margin separating first and second place in recent MaxSAT Evaluation tracks.

While minor regressions were observed in isolated cases (e.g., NuWLS on 2022), the consistent overall improvements demonstrate the utility of SGATs in guiding local search. These gains suggest that SGATs effectively prune the search space by providing better starting points.

Figure 7 illustrates these results as cactus plots, showing the number of benchmark instances solved across a range of score thresholds. SGAT-enhanced solvers consistently improve upon their vanilla counterparts under both 60s and 300s timeouts.

## D    Additional Experimental Details

All code for experiments can be found in: `https://github.com/sotam2369/SGAT-MS`

**Evaluation Environment.**    All experiments were done on a machine with AMD Ryzen Threadripper PRO 3975WX 32-Cores and two NVIDIA RTX A6000 GPUs.

### D.1    Benchmark Instances

Table 4: The number of benchmark instances used from each year of the MaxSAT evaluations

| Year | Unweighted | Weighted |
|------|------------|----------|
| MaxSAT 2020 | 75 | 57 |
| MaxSAT 2021 | 55 | 60 |
| MaxSAT 2022 | 50 | 36 |
| MaxSAT 2023 | 30 | 49 |
| MaxSAT 2024 | 39 | 34 |

We particularly performed experiments on the non-partial weighted and unweighted benchmark instances that were used throughout each year's competitions. The number of instances per year is shown in Table 4. The train/test splits for MaxSAT 2018 is given in the repository.

Table 5: Effect of number of SGAT blocks on Weighted Satisfaction Ratio (WSR).

| Metric | 1 | 2 | 3 | 4 | 5 | 6 | 7 | 8 | 9 | 10 |
|--------|-------|-------|-------|-------|-------|-------|-------|-------|-------|-------|
| WSR | 83.57 | 83.78 | 84.20 | 84.29 | 84.04 | 84.38 | 84.46 | 84.42 | 84.52 | 84.56 |

## D.2 Effect of number of layers

The number of SGAT blocks used in the model was chosen empirically. While increasing the number of blocks tends to improve model performance during training as shown in Table 5, larger models increase the runtime of the optimization procedure (Algorithm 1) and require more memory. After sweeping the number of blocks we found that performance improved up to a point and then leveled off or degraded due to these practical costs. We therefore use 6 blocks as our default, which we found to be the best trade-off between performance and resource cost in our experiments.

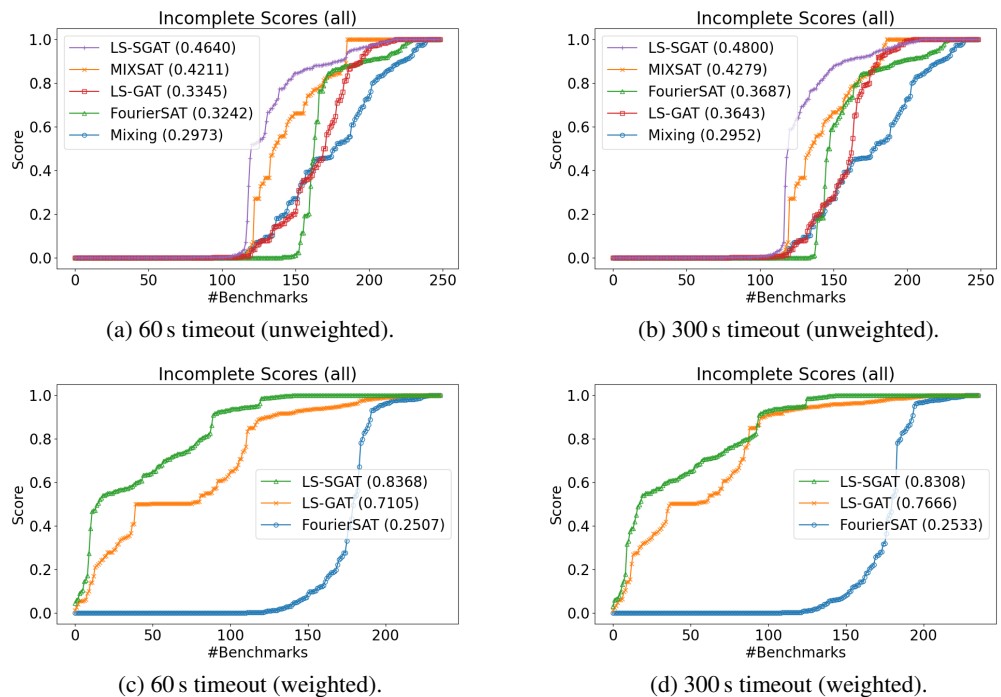

(a) 60 s timeout (unweighted).

(b) 300 s timeout (unweighted).

(c) 60 s timeout (weighted).

(d) 300 s timeout (weighted).

Figure 8: Cactus plots of *continuous* MaxSAT solvers. The $x$-axis shows the number of benchmark instances, and the $y$-axis shows the incomplete scores achieved. LS-SGAT and LS-GAT consistently outperform classical solvers such as FourierSAT, and this advantage holds across both unweighted and weighted benchmarks.

## D.3 Cactus Plots of Continuous Solvers

Figures 8a, 8b, 8c and 8d present cactus plots that compare continuous MaxSAT solvers on both unweighted and weighted benchmarks. These results focus exclusively on continuous approaches such as LS-SGAT, LS-GAT, FourierSAT, MIXSAT, and Mixing. LS-SGAT consistently achieves the best performance across all settings, with LS-GAT also outperforming traditional baselines. These trends highlight the superior performance and robustness of SGAT-based methods in continuous optimization frameworks for MaxSAT.

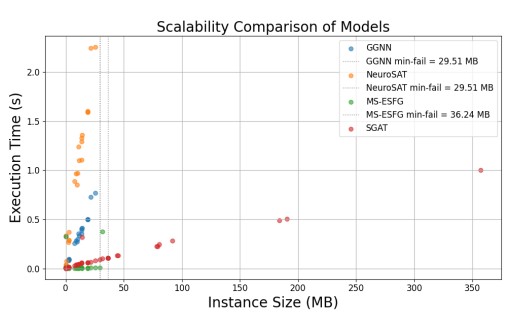

(a) Scalability of SGATs on increasing problem sizes compared to neural-based architectures.

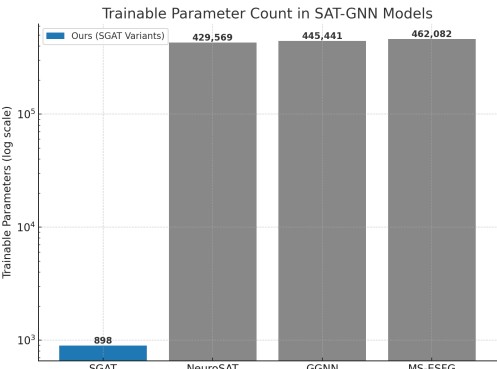

(b) Parameter count comparison across architectures.

Figure 9: Comparison of SGAT scalability and parameter efficiency.

## D.4 Scalability Analysis

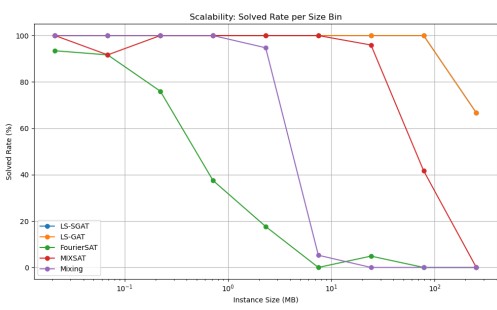

(a) Solved-rate vs. size (unweighted instances).

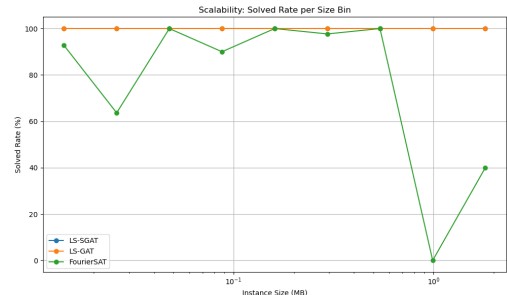

(b) Solved-rate vs. size (weighted instances).

Figure 10: Scalability analysis *for continuous MaxSAT solvers* based on the proportion of instances solved within logarithmic size bins. A point at $100\,\mathrm{MB}$ on the $y$-axis means the solver succeeded on **all** test instances whose DIMACS file is between 64MB and 128MB. LS-SGAT retains a 100% solve ratio across every bin, while LS-GAT remains competitive up to $\sim 200$MB. Classical continuous baselines (FourierSAT, MIXSAT, Mixing) degrade sharply beyond a few megabytes.

Figure 9 compares the scalability and parameter efficiency of SGATs with prior neural SAT solvers. The right subfigure shows that SGAT uses orders of magnitude fewer parameters compared to NeuroSAT, GGNN, and MS-ESFG (with only 898 parameters). This reflects the architectural simplicity of SGATs, which rely on fixed attention structures and minimal trainable components.

In contrast, prior models utilize deep multi-layer networks with complex gating or transformer-based components, leading to parameter counts exceeding 400K. Even with orders-of-magnitude fewer parameters, the left subfigure shows that SGAT maintains superior scalability: execution time grows slowly with problem size, and SGAT is the only model that remains tractable on problems exceeding 100MB. Models like NeuroSAT and GGNN begin to fail around 30MB, while MS-ESFG fails around 36MB.

These results highlight SGAT's suitability for large-scale SAT solving, where both inference speed and memory footprint are critical. By combining low parameter count with favorable runtime scaling, SGATs enable efficient reasoning on real-world datasets that challenge conventional architectures.

Figure 10 complements this view by showing the *solved-rate* curves: LS-SGAT maintains a near 100% solve ratio across every size bin, highlighting its robustness on very large weighted and unweighted instances, compared to continuous baselines like FourierSAT, MIXSAT, and Mixing.

**Algorithm 4:** Polarity-Majority Assignment

---

**Input:** CNF $F = \{C_1, \ldots, C_m\}$ with $|C_j| = k$; variables $x_1, \ldots, x_n$
**Output:** $\hat{x} \in \{0, 1\}^n$
// Count positive/negative occurrences per variable
1 **for** $i = 1, \ldots, n$ **do**
2 $\quad$ $\mathrm{pos}_i \leftarrow 0, \mathrm{neg}_i \leftarrow 0$
3 **end**
4 **for** $j = 1, \ldots, m$ **do**
5 $\quad$ **foreach** *literal* $\ell \in C_j$ **do**
6 $\quad\quad$ **if** $\ell = x_i$ **then** $\mathrm{pos}_i \leftarrow \mathrm{pos}_i + 1$
7 $\quad\quad$ **if** $\ell = \neg x_i$ **then** $\mathrm{neg}_i \leftarrow \mathrm{neg}_i + 1$
8 $\quad$ **end**
9 **end**
// Assign by literal majority
10 **for** $i = 1, \ldots, n$ **do**
11 $\quad$ **if** $pos_i \geq neg_i$ **then**
12 $\quad\quad$ $\hat{x}_i \leftarrow 1$
13 $\quad$ **else**
14 $\quad\quad$ $\hat{x}_i \leftarrow 0$
15 $\quad$ **end**
16 **end**
17 **return** $\hat{x}$

---

## E    Theoretical Analysis of SGATs

We show that a one-layer SGAT with fixed parameters realizes the polarity-majority assignment algorithm on unweighted Max-E$k$SAT and therefore attains a deterministic $\frac{1}{2}$-approximation.

**Theorem 1** (Polarity-majority assignment achieves a half-approximation on Max-E$k$SAT)**.** *Let $F = \{C_1, \ldots, C_m\}$ be a $k$-CNF formula (every clause has exactly $k$ literals). Then, the assignment produced by Algorithm 4 satisfies at least $m/2$ clauses.*

*Proof.* Give each literal weight 1 so that every clause carries total mass $k$. Let $\mathrm{pos}_i$ and $\mathrm{neg}_i$ be the positive and negative occurrence counts tracked by Algorithm 4, and define

$$W^+ = \sum_i \max\{\mathrm{pos}_i, \mathrm{neg}_i\}, \qquad W^- = \sum_i \min\{\mathrm{pos}_i, \mathrm{neg}_i\}.$$

By construction $W^+ \geq W^-$. Moreover, for each variable $x_i$, the term $\max\{\mathrm{pos}_i, \mathrm{neg}_i\} + \min\{\mathrm{pos}_i, \mathrm{neg}_i\}$ simplifies to $\mathrm{pos}_i + \mathrm{neg}_i$. Summing over $i$ therefore gives

$$W^+ + W^- = \sum_i (\mathrm{pos}_i + \mathrm{neg}_i) = mk,$$

because every clause contributes exactly $k$ literal occurrences. For instance, if a variable appears twice positively and once negatively, it contributes $2 + 1 = 3$ units of weight in total, matching the three literals that mention it.

Let $U$ be the number of unsatisfied clauses under the assignment returned by Algorithm 4. In any unsatisfied clause, each literal disagrees with the assignment on its variable. Since the algorithm picks the majority polarity for each variable, disagreeing literals belong to the minority side and therefore each contributes 1 to $W^-$. Consequently $W^- \geq kU$. Combining this with $W^+ \geq W^-$ and $W^+ + W^- = mk$ yields

$$kU \leq W^- \leq \frac{W^+ + W^-}{2} = \frac{mk}{2},$$

so $U \leq m/2$. Hence at least $m - U \geq m/2$ clauses are satisfied. $\qquad\square$

**Theorem 2** (One-block SGAT realizes Algorithm 4). *There exists SGATs with one SGAT block, Gödel t-norm, one attention head, and hidden dimension $d=1$ (scalars on nodes), using fixed parameters, that produces exactly the assignment of Algorithm 4 on any unweighted Max-EkSAT instance.*

*Proof.* We employ the initializer module to output $v^{(0)}(x_i) = \frac{1}{2}$ for every variable. With the Gödel t-norm $T_G$ computing clause valuations, each clause obtains $v_G(C_j) = \frac{1}{2}$. The SGAT update message $m_{ij}^U = v^{(0)}(x_i) \pm \left(1 - v_G(C_j)\right)$ therefore yields $m_{ij}^U = 1$ on positive literal edges and $m_{ij}^U = 0$ on negative ones. Using a single attention head with all parameters set to zero makes the variable-side softmax uniform; with hidden dimension $d=1$, node features are scalars and the post-aggregation valuation becomes $v^{(1)}(x_i) = \texttt{pos}_i/(\texttt{pos}_i + \texttt{neg}_i)$. We then directly threshold $v^{(1)}(x_i)$ at $1/2$ to set $x_i = 1$ iff $\texttt{pos}_i \geq \texttt{neg}_i$. This matches Algorithm 4 exactly. $\qquad\square$

### E.1 Polarity-Majority Assignment (PMA) vs SGAT Predictions

To better understand how the Polarity-Majority Assignment (PMA) algorithm performs in comparison with the learned SGAT predictor, we evaluated both on the unweighted benchmark sets from MaxSAT evaluations (2020–2024). For each year we report: the win counts (how many problems each method achieved a strictly higher clause satisfaction on), and the mean clause satisfaction across the test set for each method.

Table 6: Summary comparison between Polarity-Majority Assignment (PMA) and learned SGAT predictions.

| Metric | Solver | 2020 | 2021 | 2022 | 2023 | 2024 |
|---|---|---|---|---|---|---|
| Clause satisfaction | PMA | 0.7113 | 0.7025 | 0.5276 | 0.8032 | 0.7742 |
| | SGAT | 0.9216 | 0.9335 | 0.9230 | 0.9862 | 0.9910 |
| Wins | PMA | 0 | 0 | 0 | 0 | 0 |
| | SGAT | 75 | 55 | 50 | 30 | 39 |

The results shown in Table 6 indicate that the learned SGAT predictor consistently outperforms the deterministic Polarity-Majority Assignment (PMA) algorithm across all evaluated years, both in per-problem wins and clause satisfaction. PMA fails to win in any benchmarks in this comparison, underscoring our claim that SGATs can at least realize the PMA algorithm.

## F Limitations

While SGATs demonstrate strong empirical performance across multiple MaxSAT benchmarks, there are several limitations that warrant discussion. First, our framework currently does not support partial MaxSAT, where some clauses must be satisfied (hard clauses). Extending SGATs to explicitly distinguish and satisfy hard constraints remains an open challenge, particularly due to the soft nature of our t-norm-based loss formulation. Second, although SGATs generalize well across diverse datasets, their performance may still depend on the diversity and representativeness of training instances. Handling highly domain-specific or adversarially structured formulas is not guaranteed. Lastly, while our theoretical results do not cover weighted MaxSAT, they apply to general (non-partial) MaxSAT without clause weights. Extending these guarantees to handle weighted clauses or partial MaxSAT settings remains an open direction.

## G Broader Impact

The proposed SGAT architecture introduces a differentiable framework for MaxSAT solving, with potential implications in both academic and applied settings. On the positive side, SGATs provide a viable tool for integrating symbolic reasoning into neural systems, enabling end-to-end training for complex constraint-driven tasks. This could benefit domains such as autonomous driving and explainable AI, where the ability to make transparent, constraint-aware decisions is critical. In particular, our work contributes to the growing field of neuro-symbolic AI, which seeks to integrate symbolic reasoning with neural learning systems for improved interpretability and performance in

safety-critical environments. However, the integration of neural reasoning into decision-making pipelines must be approached with caution. As our method does not enforce global optimality or fairness constraints, its deployment in sensitive applications—such as law, finance, or healthcare—should be carefully monitored. Moreover, as with many machine learning systems, there is a risk that SGATs inherit biases from their training data. Transparency and rigorous evaluation remain essential for safe use.

