# OpenReview forum: "Graph-Based Attention for Differentiable MaxSAT Solving"
_NeurIPS.cc/2025/Conference — NeurIPS 2025 spotlight_

### Official Review · Reviewer_y3nh · 2025-06-20

**Clarity:** 3
**Significance:** 3
**Originality:** 3
**Rating:** 4
**Confidence:** 3

**Summary:**

This paper presents SAT-based Graph Attention Networks (SGATs), a novel class of Graph Neural Networks (GNNs) designed to solve (weighted) MaxSAT problems via continuous optimization. SGATs integrate t-norm based attention and message-passing mechanisms to approximate greedy distributed local search. A local search solver, LS-GNN, is developed to leverage SGATs iteratively.

**Questions:**

1. Why were the ontinuous optimization baselines restricted to pre-2020 methods? Have more recent MaxSAT solvers been considered?

2. While the ablation studies on MS2018 are informative, could the authors clarify whether similar analyses were attempted on more recent benchmarks? It would help understand whether the observed component-wise benefits generalize beyond the training domain.

**Ethical Concerns:**

["NO or VERY MINOR ethics concerns only"]

**Final Justification:**

The authors have clarified my comments, so I maintain my positive rating.

**Limitations:**

Yes.

**Paper Formatting Concerns:**

N/A.

**Quality:**

3

**Strengths And Weaknesses:**

Strengths:

+ The local search framework LS-GNN is simple yet effective, and fits naturally with the design of SGATs.
+ The paper includes ablation studies and theoretical analysis (approximation ratio) that enhance its technical credibility.

Weaknesses:

+ The continuous optimization baselines (Mixing from 2019, MIXSAT from 2017, FourierSAT from 2020) are relatively outdated. The paper would be more convincing if compared against more recent local search-based MaxSAT solvers.
+ While the proposed SGAT architecture is novel, the motivation for using GNNs over conventional solvers could be made more explicit.

---

> ### Author Rebuttal · Authors · 2025-07-30
>
> We thank the reviewer for positive comments regarding the theoretical findings and simple algorithm, as well as suggestions for improving the clarity of our paper. We have carefully addressed the concerns raised in the review as follows:
>
> > [W1.1/Q1] The continuous optimization baselines (Mixing from 2019, MIXSAT from 2017, FourierSAT from 2020) are relatively outdated.
>
> Regarding the continuous optimization baselines, the solvers raised in the paper are the related works that we were able to find open source implementations for.
> As for FourierSAT, newer versions were presented more recently, but we opted to use the original version as we found that newer implementations were not adaptable to the weighted MaxSAT setting. As having support for weighted MaxSAT is extremely important for performing comparisons, we have opted to use the original FourierSAT solver, the only version that supports weighted MaxSAT.
>
>
> > [W1.2/Q1] The paper would be more convincing if compared against more recent local search-based MaxSAT solvers.
>
> Due to the limit in space, we were not able to include full comparisons with state-of-the-art local search solvers in the main text.
> However, as stated in L306 and Appendix C, we have investigated the effect of using SGATs as initialization heuristics for state-of-the-art incomplete MaxSAT solvers. The results in Table 3 show that we constantly outperform the variant without heuristics, achieving state-of-the-art results on benchmark instances from 2020 to 2024, even with the overhead of producing SGAT predictions. These results highlight the power of SGATs in providing high-quality approximations for MaxSAT problems, and we feel that this is a key contribution of our work.
>
>
> > [W2] While the proposed SGAT architecture is novel, the motivation for using GNNs over conventional solvers could be made more explicit.
>
> Currently, our main goal is to provide methods that complement conventional solvers, and not to replace them. We state below two motivating reasons behind the use of GNNs (or neural networks in general):
>
> 1. **As heuristics**: While there exists highly tuned solvers that can solve MaxSAT problems efficiently, these solvers often rely on the use of heuristics that require expert knowledge. In such scenarios, neural models such as SGATs can be used to provide heuristics for performance increase, effectively reducing the cost of hand crafting these heuristics.
> 1. **For neuro-symbolic integration**: In the field of neuro-symbolic research, there is a general need for symbolic solvers that align well with continuous outputs of neural models. Solving models like ours could be used in such situations, making it crucial towards developing high performing models that align well with solvers.
>
> Lastly, as shown by many existing works, SAT problems can be naturally represented as graphs (as shown in Figure 6), supporting our choice of using GNNs over ordinary neural networks for MaxSAT. This choice is further supported in many of SAT related works, which have also shown GNNs to be effective as heuristics, supporting our first point above.
>
>
>
>
> > [Q2] While the ablation studies on MS2018 are informative, could the authors clarify whether similar analyses were attempted on more recent benchmarks? It would help understand whether the observed component-wise benefits generalize beyond the training domain.
>
> Yes. From our analysis, we have observed that our model does indeed perform similarly on more recent benchmarks, when performing similar ablation studies. While there are differences in the overall clause satisfaction, the tendency of the results closely match those shown in Figure 5, ensuring that the model performs well regardless of the dataset used.

---

> > ### Comment · Reviewer_y3nh · 2025-08-03
> >
> > The authors have clarified my comments. Since my rating has already been positive, I would like to maintain my rating.

---

> > > ### Author Response · Authors · 2025-08-04
> > >
> > > Dear Reviewer y3nh,
> > >
> > > We would like to thank the reviewer for taking the time to read our responses.
> > > We are glad to have clarified the raised points, and we appreciate the reviewer for keeping a positive score.

---

### Official Review · Reviewer_nviw · 2025-06-26

**Clarity:** 4
**Significance:** 4
**Originality:** 3
**Rating:** 6
**Confidence:** 4

**Summary:**

This paper proposes SGAT, a novel Graph Neural Network architecture, designed specifically to address (Weighted) MaxSAT problems. SGAT incorporates a t-norm-based attention and message passing mechanism that approximates greedy distributed local search in a differentiable framework. The authors also introduce LS-GNN, a local search solver leveraging SGATs. The proposed method significantly outperforms existing GNN architectures and continuous optimisation baselines such as Mixing, MIXSAT, and FourierSAT on multiple MaxSAT benchmarks. Additional experiments demonstrate the importance of SGAT's components, and highlight its potential to provide high-quality initial assignments for integration with existing SOTA solvers.

**Questions:**

- In Algorithm 1, is the local search within the while loop executed in parallel? Is it possible to assign multiple random assignments to $\alpha$ simultaneously, conduct local search in parallel, and then choose the one with the best cost? If the process executes with a single $\alpha$, is parallelisation feasible, and could it lead to performance improvements?
- In Figure 4, the test performance is generally better when models are trained on the synthetic dataset SR(U(40,200)) rather than the benchmark MS2018(2MB). This seems counter-intuitive. Could the authors provide an explanation for this phenomenon?
- In the formula for Average Clause Satisfaction (Line 244), why is the round function applied to the clause satisfaction value? It seems possible to compute a meaningful average without rounding. What is the rationale for this choice?
- In Table 1, some baseline solvers perform worse under a 300-second timeout than under a 60-second timeout in certain years. Could the authors explain why longer time limits would degrade performance in these cases?

**Ethical Concerns:**

["NO or VERY MINOR ethics concerns only"]

**Final Justification:**

The rebuttal has effectively addressed most of my questions. The authors have clarified experimental concerns and ambiguities in the algorithm, and I am confident that these will be revised accordingly. I adjust the rating, and look forward to reading the updated version of the paper.

**Limitations:**

The main limitations of this work are those already acknowledged by the authors. First, SGAT has not yet been extended to handle partial MaxSAT problems, where satisfying hard clauses is mandatory. Second, while the method shows strong performance on standard MaxSAT benchmarks, it has not been tested in broader neuro-symbolic reasoning tasks or real-world domains beyond SAT-style formulations. These limitations are rather natural next steps for future exploration. Given the flexibility of the proposed architecture and its compatibility with existing solvers, these extensions seem feasible and promising.

**Paper Formatting Concerns:**

There is no formatting issues in this paper.

**Quality:**

3

**Strengths And Weaknesses:**

Strengths
---
- The definitions of the target problems (SAT, MaxSAT, Weighted MaxSAT) are clearly and precisely explained. The mathematical formulations provided throughout the paper are well-defined, making the paper highly readable and accessible.
- The proposed GAT-based architecture, SGAT, is systematically well-structured and clearly explained. The roles of all variables in the equations are explicitly defined, and the message passing mechanism and optimisation objectives are well-formulated.
- SGAT significantly outperforms baseline GNN architectures such as NeuroSAT, GGNN, and MS-ESFG in experimental results. Moreover, the LS-GNN algorithm, built on SGAT, exhibits superior performance compared to continuous MaxSAT solver baselines like Mixing, MIXSAT, and FourierSAT.
- A well-constructed ablation study clearly shows that each component of SGAT (SGAT Layer, T-norm Layer, T-norm type) plays a crucial role in its high performance.
- SGAT demonstrates practical utility when used as an initialisation heuristic for existing SOTA Weighted MaxSAT solvers. The experimental report suggests that integrating SGAT can improve solver performance, indicating the high potential of SGAT in real-world solver integration.

Weaknesses
---
- In Algorithm 1 (LS-GNN), it is unclear how $\Delta$ in Step 18 is defined. A brief explanation of "early stopping" variable would be appreciated. In the while loop, $\alpha$ is repeatedly re-initialised with a random assignment in Step 4, and it remains unsure to me how the update in Step 18 is contributing to the process.
- In Lines 224–225, the paper mentions an analysis of the approximation ratio of SGATs. It would be also appreciated to provide some concrete numbers or at least a summary (better or worse than prior methods) regarding this ratio, even if full theoretical guarantees are difficult.

---

> ### Author Rebuttal · Authors · 2025-07-30
>
> We thank the reviewer for positive comments regarding the novelty and motivation of our SGAT model, as well as strong experimental results. We also thank the reviewer for suggestions to improve the clarity of our paper, and have carefully addressed the concerns raised in the review as follows:
>
>
> > [W1.1] In Algorithm 1 (LS-GNN), it is unclear how $\Delta$ in Step 18 is defined. Also, a brief explanation of "early stopping" variable would be appreciated.
>
> We apologize for not being able to include a full explanation of the parameters associated with the algorithm in the main text due to space constraints.
> As described in Appendix A.2, $\Delta$ is a set of randomly generated values in the range $[0, 1]$, used to essentially replace (break) part of the current assignment with a random assignment.
> Similarly, the early stopping variable is a parameter for deciding when to stop the continuous optimization process, and restart from a different assignment.
> We will make sure to include brief explanations of these parameters in the main text, and hope that this will help clarify the algorithm.
>
>
> > [W1.2] In the while loop, $\alpha$ is repeatedly re-initialised with a random assignment in Step 4, and it remains unsure to me how the update in Step 18 is contributing to the process.
>
> We thank the reviewer for pointing out this issue. The random assignment of $\alpha$ should only be done once before the while loop starts, i.e, before Step 2. We will make sure to revise the algorithm, and we apologize for the confusion caused by this mistake.
>
>
> > [W2] In Lines 224–225, the paper mentions an analysis of the approximation ratio of SGATs. It would be also appreciated to provide some concrete numbers or at least a summary (better or worse than prior methods) regarding this ratio, even if full theoretical guarantees are difficult.
>
> Due to the limit in space, we were not able to include a detailed discussion of the theoretical results in Section 4.6, and thus most of the results are shown in Appendix E.
> From our analysis, we have found that our model has capabilities of learning a solving algorithm that can guarantee satisfying at least 1/2 of the maximum satisfiable clauses. We will make sure to revise the paper to include the findings as well as a brief discussion of the theoretical results in Section 4.6, and hope that this will help clarify the significance of this result.
>
>
> > [Q1] In Algorithm 1, is the local search within the while loop executed in parallel? Is it possible to assign multiple random assignments to $\alpha$ simultaneously, conduct local search in parallel, and then choose the one with the best cost? If the process executes with a single $\alpha$, is parallelisation feasible, and could it lead to performance improvements?
>
> Implementation-wise, yes. We have already supported partial parallelization, by simply passing multiple instances of the problem and assignment batched together.
> However, as this requires a single model to find solutions from multiple different initial assignments, we have observed that there is a clear trade-off between efficiency and performance.
> Nevertheless, this is a very under-explored area of our work, and we will make sure to include a discussion of this in the paper, as well as refer to this as future work on this topic.
>
>
> > [Q2] In Figure 4, the test performance is generally better when models are trained on the synthetic dataset SR(U(40,200)) rather than the benchmark MS2018(2MB). This seems counter-intuitive. Could the authors provide an explanation for this phenomenon?
>
> We agree that this is an interesting phenomenon. We think that one reason for this occuring, is that the MS2018(2MB) dataset contains large instances, leading us to use a batch size of 1 during training for models other than SGATs. This leads to the model not being able to learn from multiple instances at once, degrading the overall performance of the model.
>
> Another reason we think this occurs, is due to the variety of problem types provided in the MS2018(2MB) dataset. As the dataset contains a wide range of structured problem types (5 types to be exact), it becomes harder to train -- especially with single-instance batches. On the other hand, our model is very lightweight, and is capable of handling larger batch sizes. We feel that this is one main factor that leads to the performance of SGATs being better than existing models on both datasets as well.
>
>
> > [Q3] In the formula for Average Clause Satisfaction (Line 244), why is the round function applied to the clause satisfaction value? It seems possible to compute a meaningful average without rounding. What is the rationale for this choice?
>
> As the reviewer has pointed out, it is possible to compute a meaningful average without rounding.
> However, in our setting, the purpose of rounding was to align the performance metric of models with how MaxSAT solvers are typically evaluated.
> As our final goal is to find a solution that satisfies as many clauses as possible with regards to weight, the formula (L244) is designed to compute the average weights of all satisfied clauses.
>
>
>
> > [Q4] In Table 1, some baseline solvers perform worse under a 300-second timeout than under a 60-second timeout in certain years. Could the authors explain why longer time limits would degrade performance in these cases?
>
> This is mainly due to how we evaluate the incomplete scores. The incomplete scores use the best scores found by all solvers, and thus if any of the other solvers in the same timeout set find a better solution, the other solvers' scores will worsen even if they had the exact same solution as before.
> However, we do agree that this may be confusing, and will make sure that the best scores are computed from all solvers regardless of timeout, so as to keep the scores comparable.

---

> > ### Comment · Area_Chair_sdMz · 2025-08-04
> >
> > Dear Reviewer, please engage into discussions with the Authors as the deadline for this key phase of the NeurIPS review process is only a couple of days away.

---

### Official Review · Reviewer_2ENR · 2025-07-01

**Clarity:** 3
**Significance:** 3
**Originality:** 3
**Rating:** 5
**Confidence:** 3

**Summary:**

The paper proposes a solver based on a graph neural network. The authors present the architecture and how the model is the use for solving MaxSAT problems.

After the model and algorithm presentation the authors show the performance on standard MaxSAT instances and how the proposed approach works with ablation study on architecture choices.

**Questions:**

It isr elated to the weakeness point: would it be possible to define a different local search? Could you elaborate? there is part in the ANNEX that extend, could you spring something in the main text?


In algorithm 1 is not clear what \alpha refers to, since there is the \alpha of the attention, but this is not a scalar

In general, it is not clear what is learned and how it is learned.

It would also be nice to see where the approach is with respect to more mature solvers: "MaxSAT Evaluation 2023: Summary of Unweighted Exact Track"
(https://maxsat-evaluations.github.io/2023/results/exact/unweighted/summary.html) or any other source.

**Ethical Concerns:**

["NO or VERY MINOR ethics concerns only"]

**Final Justification:**

The authors left out some important information on the method (which parameters are trained in particular, and the distinction of the training and testing phases), but the approach is sound and the performance promising.

I am keeping my score.

**Limitations:**

yes, in the annex

**Paper Formatting Concerns:**

no found

**Quality:**

3

**Strengths And Weaknesses:**

Strengths
1. clear presentation of the architecture and the separation of the role of the network and the algorithm
2. Experiments that highlight the performance and the role of the architecture choices.

I personally like that the bipartite graph, the transformation from variable to clause is base on the T-norm.

Weaknesses
1. I would have like to see the analysis with other heuristics

---

> ### Author Rebuttal · Authors · 2025-07-30
>
> We thank the reviewer for positive comments regarding the novelty and motivation of our SGAT model, as well as on our experimental results. We also thank the reviewer for suggestions to improve the clarity of our paper, and have carefully addressed the concerns raised in the review as follows:
>
> > [W1] I would have liked to see analysis with other heuristics
>
> Currently, we have already have done multiple comparisons with existing heuristics. Firstly, in L306, Appendix C, we have proposed to integrate our model as initialization heuristics for multiple state-of-the-art local search solvers, and compared them to the original solvers' initialization mechanisms.
> The results in table 3 shows that our proposed method outperforms the original, leading to state-of-the-art results on benchmark instances from 2020 to 2024.
> These results indicate that our model can be effectively used as heuristics for local search solvers, and even achieve state-of-the-art results on benchmark instances.
>
> Furthermore, we can regard the Mixing method as a heuristic. The MIXSAT solvers uses the Mixing method as a heuristic, performing branch-and-bound to find solutions for MaxSAT problems. As we had compared to this approach, we can essentially claim that our algorithm is being compared with existing heuristic methods, and outperforming them.
>
> In terms of direct comparisons with algorithms, we have results comparing our model with the CBLM algorithm, a novel algorithm with theoretical guarantees that we show is learnable by the SGAT model (Appendix E). The results shown in the table below show how many instances each solver was able to satisfy more clauses than the other on, with SGATs being used in a one-shot prediction setting.
> The results show that our model is able to outperform CBLM on almost all (98%) benchmark instances, demonstrating that our model is as good as CBLM in terms of learning to solve MaxSAT problems.
>
> |Clause Satisfaction Win Count| 2020 | 2021 | 2022 | 2023 | 2024 | Total |
> |----------|------|------|------|------|------|------|
> | **CBLM** |   1  |   1  |   1  |  1   |  1   | 5   |
> | **SGAT** |  92  |  54  |  49  | 29   | 38   | 262 |
>
>
> Finally, we feel that the comparison of our model with more prominent heuristics such as variable selection heuristics in local search algorithms are a promising direction for future work. We will make sure to include a brief discussion of this as future work, if accepted.
>
>
> > [Q1] It is related to the weakeness point: would it be possible to define a different local search? Could you elaborate? there is part in the ANNEX that extend, could you spring something in the main text?
>
> Yes, a different local search algorithm to Algorithm 1 definitely can be defined on top of SGATs. One clear method is as initialization heuristics for existing solvers (Algorithm 2,3), as we have described in `[W1]`. While the reliance of SGATs in this method is lower, the positive results demonstrate its potential in various settings and algorithms.
> Furthermore, we feel that the development of such algorithms is a promising direction for future research, and so we will make sure to include a brief discussion of this in the final version of the paper, if accepted.
>
>
> > [Q2] In Algorithm 1, it is not clear what $\alpha$ refers to, since there is the $\alpha$ of the attention
>
> In Algorithm 1, $\alpha$ refers to the current assignment of the variables, a standard notation in the literature of MaxSAT solving. On the other hand, $\alpha$ introduced in Section 3.2 refers to the attention values, which is also the standard notation used in works regarding attention for GNNs. However, we do agree that this may be confusing, and so we will make sure to either rename the variable in Algorithm 1 or Section 3.2, or clarify the notation when the variables are introduced.
>
>
> > [Q3] In general, it is not clear what is learned and how it is learned.
>
> During pretraining, the model learns in general what types of clauses the variables should focus on when solving any type of MaxSAT problem, and during testing, the model refines this knowledge to solve specific instances.
> Intuitively, we can regard this as learning a general heuristic that can be applied to a variety of MaxSAT problems, and then refining this heuristic to solve a specific problem instance.
>
> Specifically, there are parameters within the model that requires learning, such as $\mathbf{a}$, $\mathbf{W}$ and $\mathbf{W}_e$ shown in the equation at L180.
> These parameters are used to compute the attention values, which in turn represents which clauses the variable should focus on satisfying.
> We can regard the parameters as heuristics for performing greedy distributed local search.
>
> In our model, the main parameters that get learned are the weights used in the attention mechanisms (such as $a$, $W$ and $W_e$ shown in the equation at L180).
> These weights are used to compute the attention values, which in turn represents which clauses the variable should focus on satisfying.
> Intuitively, attention values represent the heuristics for greedy distributed local search that this model approximates, indicating that our model essentially learns heuristics for solving MaxSAT problems.
>
> As for how we train, we first convert problems into graphs, then predict the assignment of the variables, and compute the loss based on the predicted assignment and the ground truth assignment. These are done on benchmark instances, as shown in Section 5.1.
>
> > [Q4] It would also be nice to see where the approach is with respect to more mature solvers: "MaxSAT Evaluation 2023: Summary of Unweighted Exact Track" (https://maxsat-evaluations.github.io/2023/results/exact/unweighted/summary.html), or any other source.
>
> As stated in `[W1]` and `[Q1]`, our model has been integrated into state-of-the-art local search based MaxSAT solvers as initialization heuristics, where we achieved state-of-the-art on benchmark instances from 2020 to 2024 (Table 3). We feel that that similar integrations can be accomplished for exact solvers as well, and that this will have similar merits as the local search solvers. We will make sure to include a discussion of this in the final version of the paper, if accepted.

---

> > ### Comment · Reviewer_2ENR · 2025-08-01
> > **Algorithm**
> >
> > Dear Authors,
> >
> > Thank you for the clarification. I hope you can effectively clarify the distinction among variables (not using the same symbols), clarify the training and test time (what is trained, which variables are used in the test that have been trained, and which variables are refined a test time), and also the role of the heuristic at test time.  A discussion of ML+Heuristics and "exact solvers" is useful to identify what still needs to be done in the area.

---

> > > ### Author Response · Authors · 2025-08-04
> > >
> > > Dear Reviewer 2ENR,
> > >
> > > We thank the reviewer for taking the time to read our responses.
> > > We are glad to have clarified some of the points raised, and we agree that these suggestions will help improve the clarity and positioning of our work.
> > > We will make sure to incorporate them in the final version of our paper if accepted.

---

### Official Review · Reviewer_xFG8 · 2025-07-01

**Clarity:** 3
**Significance:** 3
**Originality:** 3
**Rating:** 4
**Confidence:** 4

**Summary:**

The authors propose SGAT, a graph attention network architecture specifically designed to solve the MaxSAT problem, which is claimed to be the first GNNs to handle weighted MaxSAT problems. The technical contributions include the t-norm based attention and SGAT layer, an approximation of greedy distributed local search. SGAT is demonstrated to achieve better solutions compared with existing GNN models on MaxSAT problems from both randomly generated and competition benchmarks. Moreover, by integrating SGAT with a local search framework, the proposed algorithm outperforms other continuous optimisation algorithms in runtime on weighted MaxSAT Evaluation benchmarks.

**Questions:**

- Has a comparison been made with traditional discrete search solvers? If so, does SGAT-based local search have any advantages?
- How was the choice made to have 6 SGAT blocks in the model, and what effects would setting more or fewer layers have on the results?
- According to my knowledge, some weighted MaxSAT benchmarks in MaxSAT Evaluations have more than 100,000 variables. How can such large graphs be loaded onto the GPUs for training?

**Ethical Concerns:**

["NO or VERY MINOR ethics concerns only"]

**Final Justification:**

Keep the score

**Limitations:**

Yes

**Paper Formatting Concerns:**

- There is a mixed use of different citation formats.
- The dot in the titles of Section 4.6 and 5.3.1 are unnecessary.

**Quality:**

3

**Strengths And Weaknesses:**

### Strengths
- The proposed SGAT is novel and efficient, which is the first GNN model to work on weighted MaxSAT instances, a critical problem of various domains in CS&AI.
- The t-norm attention and SGAT layer are well-motivated, and ablation studies have demonstrated their effectiveness and stability.
- Experiments on MaxSAT Evaluation benchmarks are promising, which have shown the ability of NN-based approach on these challenging practical problems.
- The manuscript is generally well-organised and easy to follow.

### Weaknesses
- Readers may find the term "t-norm attention" confusing, as it is not explained until Section 4.2. Explaining it when first mentioned would improve the clarity.
- It should be clarified that Algorithm 1 can solve weighted MaxSAT instances to strengthen the contribution.
- There is no meaningful conclusion from Section 4.6.

---

> ### Author Rebuttal · Authors · 2025-07-30
>
> We thank the reviewer for positive comments regarding the novelty and motivation of our SGAT model, as well as on our experimental results. We also thank the reviewer for suggestions to improve the clarity of our paper, and have carefully addressed the concerns raised in the review as follows:
>
> > [W1] Readers may find the term ”t-norm attention” confusing, as it is not explained until Section 4.2. Explaining it when first mentioned would improve the clarity.
>
> While we currently do have a brief explanation of how we incorporate t-norms and attention mechanisms in our model in L38, we agree that the term "t-norm based attention" may be confusing to the general reader.
> We will make sure to clarify the connection, by rewriting the sentence at L38 to:
>
>
> "SGATs are composed of GNN layers with novel t-norm based attention, which are attention mechanisms that operate on values computed using t-norms."
>
>
> > [W2] It should be clarified that Algorithm 1 can solve weighted MaxSAT instances to strengthen the contribution.
>
> We thank the reviewer for pointing this out. We completely agree that this is a key contribution of our work to clarify, and will make sure to revise the paper to explicitly state that Algorithm 1 can be applied to weighted MaxSAT instances.
>
>
> > [W3] There is no meaningful conclusion from Section 4.6.
>
> Due to the limit in space, we were not able to include a detailed discussion of the theoretical results in Section 4.6, and thus most of the results are shown in Appendix E.
> In this section, we have theoretically analyzed that our model has capabilities of learning a solving algorithm that can guarantee satisfying at least 1/2 of the maximum satisfiable clauses, which we believe to be meaningful.
> We will make sure to revise the paper to include a brief discussion of the theoretical results in Section 4.6, and hope that this will help clarify the significance of our theoretical results.
>
>
> > [Q1] Has a comparison been made with traditional discrete search solvers? If so, does SGAT-based local search have any advantages?
>
> Yes, we have made comparisons with traditional discrete search solvers.
> However, due to space constraints, we were not able to include full comparisons in the main text.
> As shown in L306 and Appendix C, we have proposed to integrate our model as initialization heuristics for multiple state-of-the-art local search solvers, and compared them to the original solvers' initialization mechanisms.
> The results in table 3 shows that our proposed method outperforms the original, achieving state-of-the-art results on benchmark instances from 2020 to 2024.
> These results indicate that our model can be effectively used as heuristics for local search solvers, making it an interesting direction for future works as well.
>
> > [Q2] How was the choice made to have 6 SGAT blocks in the model, and what effects would setting more or fewer layers have on the results?
>
> In our experiments, the number of blocks were decided empirically.
> While increasing the number of blocks lead to an increase in model performance during training, having large models will make the optimization procedure in Algorithm 1 take longer, as well as require more memory.
> As this generally leads to degrading performance, we have decided to use 6 blocks, the setting that we found to be optimal.
>
> | Blocks | 1     | 2     | 3     | 4     | 5     | 6     | 7     | 8     | 9     | 10    |
> |--------|-------|-------|-------|-------|-------|-------|-------|-------|-------|-------|
> | Average Clause Satisfaction | 83.57 | 83.78 | 84.20 | 84.29 | 84.04 | 84.38 | 84.46 | 84.42 | 84.52 | 84.56 |
>
>
> > [Q3] According to my knowledge, some weighted MaxSAT benchmarks in MaxSAT Evaluations have more than 100,000 variables. How can such large graphs be loaded onto the GPUs for training?
>
> We confirm that some instances have up to a million variables, and these huge instances are indeed not usable during training. Therefore, during our training phase, we only use instances that have file sizes below 2MB as stated in L243. Furthermore, for existing models, we set a batch size of 1, as larger batch sizes were not feasible even in this setting. In contrast, our model uses batch sizes of 4, which was possible due to the lightweight nature of our model, which we have discussed in Appendix D.4.
>
> > Formatting concerns of mixed citation formats, and dots in titles of sections where unnecessary.
>
> We thank the reviewer for pointing out the formatting issues, and will make sure to address them in the final version of the paper if accepted.

---

> ### Comment · Reviewer_xFG8 · 2025-08-03
>
> I appreciate the author’s carefully answering the questions I raised. I will keep my current score

---

> > ### Author Response · Authors · 2025-08-04
> >
> > Dear Reviewer xFG8,
> >
> > We would like to thank the reviewer for taking the time to read our responses and keeping a positive score.

---

### Official Review · Reviewer_E9ST · 2025-07-02

**Clarity:** 3
**Significance:** 2
**Originality:** 3
**Rating:** 4
**Confidence:** 4

**Summary:**

This paper introduces a new neural network for solving the (weighted) MaxSAT problem. Having converted a MaxSAT problem to an appropriate graph representation, the architecture consists of 3 parts: (i) T-norm Aggregation, (ii) SGAT layers, (iii) A normalization layer. The model is then be used in a local search framework to complete the approach. The authors include ablations to confirm their design choice of using SGAT (vs GAT) and the T-norm and for testing different T-norms. The authors provide results on two MaxSAT datasets, comparing the approach with previous SAT or MaxSAT architectures. Their model produces favorable results.

**Questions:**

Please refer to the strengths and weaknesses.

**Ethical Concerns:**

["NO or VERY MINOR ethics concerns only"]

**Final Justification:**

My main concerns were resolved, hence the positive score.

**Limitations:**

I think the limitations could be discussed in more detail - please see weaknesses.

**Paper Formatting Concerns:**

None noticed.

**Quality:**

3

**Strengths And Weaknesses:**

Strengths
* Favorable results when compared with existing neural network SAT solvers
* Nice ablations

Weaknesses
* Contribution: Like several other papers in this direction of research (combinatorial optimization with neural networks), I feel this work does not convince me that using neural networks is necessary or even helpful here. The proposed approach aligns very closely with greedy distributed local search (as clearly stated by the authors - which I appreciate), and I am not sure what it contributes on top:
* So firstly: CBLM and an appropriate greedy local search should be included as a baseline. The proposed model should outperform these baselines to justify itself.
* Secondly: As far as I can tell with this approach there is not much transfer of knowledge between training and testing. Each problem instance essentially has to be solved separately. Perhaps the authors could elaborate on what knowledge they expect the model weights to be able to learn which can be transferred between problem instances. I would appreciate some intuition here to motivate the approach. To stress this point further, the update messages passed between nodes represent the value in which the clause wants the variable to be updated, rather than some more complex embeddings that somehow represent the structure/topology of the problem instance, which could be transferable between problem instances. I'm happy to receive pushback on this if the authors do not agree with my intuition.

Questions
* Is the average clause satisfaction the standard metric here? I would have expected that the variables would be rounded (binarized) first, and then the (weighted) number of satisfied clauses calculated.

---

> ### Author Rebuttal · Authors · 2025-07-29
>
> We thank the reviewer for positive comments regarding our results and ablations, as well as for insightful comments and suggestions to improve our paper. We have carefully addressed the concerns raised in the review as follows:
>
> > [W1] The proposed approach aligns very closely with greedy distributed local search (as clearly stated by the authors - which I appreciate), and I am not sure what it contributes on top.
>
> Our model has been developed by aligning with greedy distributed local search, and based on it the proposed approach has come with a number of contributions and advantages:
> - **Novelty**: To learn solving MaxSAT problems, we employ a novel GNN with attention for greedy distributed local search, an approach that has never been proposed before in the literature.
> - **Performance**: As shown with multiple experiments, our model outperforms existing neural models, showing that it can effectively learn how to solve MaxSAT problems in general, as opposed to overfitting to a specific problem type. Furthermore, when used with our proposed local search algorithm, our model outperforms existing (non-neural) continuous optimization baselines, demonstrating our model's efficiency in continuous MaxSAT solving.
> - **Scalability**: Even though we were able to gain high performance, our model architecture is a still a fraction of the size of existing models (Appendix D.4). As scalability is generally a bottleneck within neuro-symbolic methods, our model can be effectively used in a variety of settings.
>
> While these are not the only advantages, we believe our proposed architecture contributes largely to multiple fields including neuro-symbolic, as well as MaxSAT solving.
>
>
>
> > [W2.1] CBLM should be included as a baseline.
>
> In the table below, we show additional results that compare our pretrained model's one-shot output performance with the CBLM algorithm on MaxSAT Evaluation benchmark instances. As the CBLM algorithm only supports non-weighted instances, we have tested on all non-weighted instances from the MaxSAT Evaluation 2020 to 2024, as shown in Table 4. The table below shows the number of instances where each solver found better solutions than the other, with SGAT being used in a one-shot prediction setting.
>
> The results show that our model finds better solutions than CBLM for _almost all (98%)_ instances. This demonstrates the superiority of our model architecture, empirically supporting our theoretical study regarding our model's capability of learning solving strategies.
> We feel that this is a strong contribution of our work, and will make sure to include these results in the final version of the paper, if space allows.
>
> |Clause Satisfaction Win Count| 2020 | 2021 | 2022 | 2023 | 2024 | Total |
> |----------|------|------|------|------|------|------|
> | **CBLM** |   1  |   1  |   1  |  1   |  1   | 5   |
> | **SGAT** |  93  |  54  |  49  | 29   | 38   | 262 |
>
>
>
> > [W2.2] Greedy local search algorithms should be included as a baseline.
>
> Due to space constraints, we were not able to include full comparisons with state-of-the-art local search solvers in the main text. However,
> as shown in L306 and Appendix C, we have proposed to integrate our model as initialization heuristics for multiple state-of-the-art local search solvers, and compared them to the original solvers' initialization mechanisms. The results in table 3 shows that our proposed method outperforms the original, leading to state-of-the-art results on benchmark instances from 2020 to 2024.
> We believe that this demonstrates the potential of our work and model in the broader context of MaxSAT solving.
>
>
>
> > [W3.1] As far as I can tell with this approach there is not much transfer of knowledge between training and testing. Each problem instance essentially has to be solved separately. Perhaps the authors could elaborate on what knowledge they expect the model weights to be able to learn which can be transferred between problem instances. I would appreciate some intuition here to motivate the approach.
>
> Firstly, we would like to clarify that our model is pretrained on dataset MS2018(2MB). Our algorithm then takes a copy of the pretrained model, and optimizes it to solve a specific problem instance, with no model parameters being shared between testing runs.
>
> During pretraining, the model learns in general what types of clauses the variables should focus on when solving any type of MaxSAT problem, and during testing, the model refines this knowledge to solve specific instances.
> Intuitively, we can regard this as learning a general heuristic that can be applied to a variety of MaxSAT problems, and then refining this heuristic to solve a specific problem instance.
>
> Specifically, there are parameters within the model that requires learning, such as $\mathbf{a}$, $\mathbf{W}$ and $\mathbf{W}_e$ shown in the equation at L180.
> These parameters are used to compute the attention values, which in turn represents which clauses the variable should focus on satisfying.
> We can regard the parameters as heuristics for performing greedy distributed local search.
>
> Empirically, we have observed a significant decrease in performance when the model is used without pretraining, further supporting our claim that knowledge transfer is indeed present.
>
> > [W3.2] To stress this point further, the update messages passed between nodes represent the value in which the clause wants the variable to be updated, rather than some more complex embeddings that somehow represent the structure/topology of the problem instance, which could be transferable between problem instances.
>
> In Graph Neural Networks (GNNs), the structure/toplogy of graphs gets represented as embeddings by the weights of the model. While our embeddings are in the form of fuzzy logic values, these too, get computed with the weights of the model. Thus, we believe that the structure/topology of the problem instance is indeed captured by our model, and that the embeddings are not simply just values for updating assignments.
>
> > [Q1] Is the average clause satisfaction the standard metric here? I would have expected that the variables would be rounded (binarized) first, and then the (weighted) number of satisfied clauses calculated.
>
> Our metric indeed agrees with the reviewer's expectation regarding how the clause satisfactions should be computed. In our current setting, we use Gödel t-norms to evaluate the satisfaction of each clause. In this case, rounding the clause values afterwards does not change the outcome to when the variables are binarized first. However, we now feel that the terminology of "average clause satisfaction" is misleading, and should be called "(average) clause weight", and so we will make sure to revise this in a final version of the paper.

---

> > ### Comment · Area_Chair_sdMz · 2025-08-04
> >
> > Dear Reviewer, please engage into discussions with the Authors as the deadline for this key phase of the NeurIPS review process is only a couple of days away.

---

> > ### Comment · Reviewer_E9ST · 2025-08-04
> >
> > Thank you for the clarifications. The rebuttal has addressed my main concerns. I will raise my score.
> >
> > Regarding the additional CBLM results, I would also like to see the average clause satisfaction scores in this comparison.

---

> > > ### Author Response · Authors · 2025-08-05
> > >
> > > Dear Reviewer E9ST,
> > >
> > > We would like to thank the reviewer for taking the time to read our responses, and are glad to have been able to clarify the main concerns.
> > > We further appreciate the reviewer for improving the score, as well as for constructive comments in raising the overall quality of the paper.
> > >
> > > Regarding the average clause satisfaction, we have obtained the results, which are shown in the table below:
> > > |Average Clause Satisfaction | 2020 | 2021 | 2022 | 2023 | 2024 | Overall |
> > > |----------|------|------|------|------|------|------|
> > > | **CBLM** | 0.74 | 0.73 | 0.57 | 0.87 | 0.82 | 0.73 |
> > > | **SGAT** | 0.92 | 0.93 | 0.92 | 0.98 | 0.99 | 0.94 |
> > >
> > > We can observe that the average clause satisfactions are much higher, showing that our model was clearly able to outperform CBLM.
> > > We will make sure to incorporate these results into the final version of the paper if accepted.

---

### Note · Authors · 2025-08-13

Dear Reviewers, ACs, and SACs,

We thank all reviewers for their constructive comments, which have helped us improve both the clarity and strength of our work.
Key points from the reviews and subsequent discussions are summarized as follows.

- **Novelty & Motivation** – SGATs are the first GNNs to handle weighted MaxSAT, with a design closely aligned to greedy distributed local search, supported by t-norm based attention and message passing.
- **Ablation Studies** – Comprehensive analysis confirms the importance of both SGAT layers and t-norm choice.
- **Strong Empirical Performance** – SGATs outperform existing neural SAT/MaxSAT models while being highly scalable due to their lightweight architecture and also outperform continuous optimization solvers when integrated into a simple yet effective local search algorithm.
- **Integration with SOTA Solvers** – When used as initialization heuristics for state-of-the-art local search solvers, SGATs consistently outperform their original versions (2020–2024 benchmarks).
- **Theoretical Contribution** – SGATs can learn the CBLM algorithm, which guarantees satisfying at least half as many clauses as the optimal solution. Empirically, with one-shot predictions, SGATs outperform CBLM on 98% of benchmark instances.

On the other hand, we have identified the following improvement points in the paper.
- **Clarity** – We will clarify the explanation of t-norm based attention earlier in the paper, and state explicitly that Algorithm 1 also applies to weighted MaxSAT. Furthermore, in Algorithm 1, we will improve the notation and correct the initialization placement.

We believe these strengths, along with the planned clarifications, make the work a strong and well-prepared submission.

---

### Decision · Program_Chairs · 2025-09-17

**Decision:**

Accept (spotlight)

**Comment:**

The submission introduces a GNN architecture for differentiable MaxSAT solving, incorporating t-norm–based attention mechanisms and message passing inspired by greedy distributed local search. A key original contribution is the design of a GNN that could be applied effectively to weighted MaxSAT.

The reviewers unanimously supported acceptance. The contributions were seen as both novel and technically sound, and the results compelling. SGATs outperform several neural and continuous optimization baselines across standard MaxSAT benchmarks, and show excellent performance when integrated into state-of-the-art local search solvers. Reviewers highlighted the clarity of the problem formulation, the soundness of the SGAT architecture, and the practical utility of using SGATs as initialization heuristics for existing solvers.

The rebuttal was effective in addressing all key concerns, including a clearer explanation of the learning process and heuristic used. Important points such as notation ambiguity, missing runtime baselines, and theoretical insights on approximation were also clarified or promised to be revised.

Reviewers encourage the authors to include missing technical details,  which had been omitted in the original version but discussed extensively in rebuttal discussion, in the final version.

I recommend to accept this paper.